# Bivariate Matrix-valued Linear Regression (BMLR): Finite-sample performance under Identifiability and Sparsity Assumptions

**Nayel Bettache**
Department of Statistics and Data Science, Cornell University,
Capital Fund Management, 23 rue de l'Université, 75007 Paris, France
nayel.bettache@cfm.com

## Abstract

This paper studies a bilinear matrix-valued regression model where both predictors and responses are matrices. For each observation $t$, the response $Y_t \in \mathbb{R}^{n \times p}$ and predictor $X_t \in \mathbb{R}^{m \times q}$ satisfy $Y_t = A^* X_t B^* + E_t$, with $A^* \in \mathbb{R}_+^{n \times m}$ (row-wise $\ell_1$-normalized), $B^* \in \mathbb{R}^{q \times p}$, and $E_t$ independent Gaussian noise matrices.

The goal is to estimate $A^*$ and $B^*$ from the observed pairs $(X_t, Y_t)$. We propose explicit, optimization-free estimators and establish non-asymptotic error bounds, including sparse settings. Simulations confirm the theoretical rates. We illustrate the practical utility of our method through an image denoising application.

## 1 Introduction

Supervised learning is a core task in modern data analysis, often applied to high-dimensional datasets. With recent advances in data acquisition technologies, many real-world datasets now exhibit intrinsic matrix structures. Examples include spatiotemporal measurements, dynamic imaging, and multivariate longitudinal studies, where rows and columns encode distinct dimensions such as time, space, or experimental conditions [8, 27]. In such settings, both the covariates and the responses may be naturally represented as matrices.

Traditional linear regression models are designed to predict a scalar outcome from a vector-valued covariate.. When the response is vector-valued, a naive approach consists in fitting a separate linear model to each coordinate. However, this strategy ignores the multivariate structure of the response. To address this, several works [6, 18, 3] propose modeling the stacked regression coefficients as a low-rank matrix, leading to the *multivariate linear regression* model. Other works have focused on predicting scalar responses from matrix-valued covariates [38, 28], leading to the *trace regression* model where the parameter becomes a matrix, usually assumed to have a low-rank structure. In our setting, both the covariates and responses are matrix-valued. It is again not appropriate to fit independent trace regression models to each entry of the output matrix. Thus, we propose a bilinear model that explicitly couples the row and column structures of the input and output matrices.

In this paper, we extend these lines of work by studying a *bivariate matrix-valued linear regression* (BMLR) model, where both the predictor and the response are matrix-valued. This framework captures richer structural dependencies and arises naturally in applications such as spatiotemporal forecasting and dynamic network modeling. The BMLR model considers $T$ independent observations $(X_t, Y_t)_{t=1}^T$, where each predictor matrix $X_t \in \mathbb{R}^{m \times q}$ and response matrix $Y_t \in \mathbb{R}^{n \times p}$ satisfy the relationship

$$Y_t = A^* X_t B^* + E_t, \quad t \in [T] := \{1, \dots, T\}. \tag{1}$$

The unknown parameter matrices are $A^* \in \mathbb{R}_+^{n \times m}$ and $B^* \in \mathbb{R}^{q \times p}$. The non-negativity constraint helps ensuring identifiability and interpretability. In many applications, $A^*$ represents mixing

weights, attention coefficients, or interaction strengths that are naturally non-negative. Note that the model in (1) remains unchanged if $A^*$ is scaled by a positive constant and $B^*$ is divided by the same constant. To resolve this identifiability issue, we impose the constraint that each row of $A^*$ has unit $\ell_1$-norm. The noise matrices $(E_t)_{t=1}^{T} \in \mathbb{R}^{n \times p}$ have independent and identically distributed entries, each drawn from a centered Gaussian distribution with variance $\sigma^2$. We discuss more general noise structures in Appendix A. The goal is to estimate the parameters $A^*$ and $B^*$ from the observed data $(X_t, Y_t)_{t=1}^{T}$. Once estimated, these parameters can be used to predict the out-of-sample response $Y_{T+1}$ given a new covariate matrix $X_{T+1} \in \mathbb{R}^{m \times q}$. This model builds on a growing literature dedicated to matrix-valued regression with matrix-valued observations [32, 23, 16, 29, 7, 11, 4], and complements recent advances in matrix autoregression and dynamic systems [9, 10, 48, 49, 24, 43, 2, 30, 41, 50]. These approaches leverage the matrix structure of the data to improve interpretability and predictive performance.

Notice that for any $t \in [T]$, $A^*$ captures the link between the rows of $Y_t$ and the rows of $X_t B^*$ while $B^*$ captures the link between the columns of $Y_t$ and the columns of $A^* X_t$. This bilinear structure encodes interactions along both matrix dimensions and is inherently richer than what standard multivariate regression models capture. If one ignores the matrix structure and vectorizes both sides, the model becomes $\mathrm{vec}(Y_t) = (B^*)^\top \otimes A^* \cdot \mathrm{vec}(X_t) + \mathrm{vec}(E_t)$, where $\mathrm{vec}(\cdot)$ denotes the column-wise vectorization of a matrix and $\otimes$ is the Kronecker product [21]. In this formulation, the regression reduces to a standard multivariate linear model with $T$ vectorized observations and coefficient matrix $M^* = (B^*)^\top \otimes A^*$, which can be estimated directly using ordinary least squares [11]. However, this vectorized approach hides the problem structure. Estimating $A^*$ and $B^*$ from an estimate $\hat{M}$ of $M^*$ amounts to solving a Kronecker factorization problem. It reduces to finding the nearest rank-one matrix in the Kronecker product space [33]. This is a non-convex problem that discards the individual matrix roles of $A^*$ and $B^*$. Moreover, the vectorized formulation leads to estimators with high variance when $nmpq \gg T$, as it fails to exploit the low-dimensional structure induced by the bilinear form.

In contrast, our approach preserves and exploits the bilinear structure directly in the matrix domain, allowing for interpretable and efficient estimation of $A^*$ and $B^*$ without solving the intractable Kronecker decomposition problem.

## 1.1 Context

Given the observations $(X_t, Y_t)_{t=1}^{T}$, the linear regression framework models the relationship between the responses and the predictors using an unknown linear map $f : \mathbb{R}^{m \times q} \to \mathbb{R}^{n \times p}$. For each $t \in [T]$ this relationship is expressed as $Y_t = f(X_t) + E_t$ where $E_t$ represents the noise term. The goal is to learn the unknown function $f$ only using the given observations. In a parametric setting we assume that $f$ belongs to a predefined class of functions.

A naive approach would be to break down the linear relationship by focusing on each entry of the responses individually. This leads to consider for $i \in [n]$ and $j \in [p]$ a linear functional $f_{ij} : \mathbb{R}^{m \times q} \to \mathbb{R}$ such that
$$[Y_t]_{ij} = f_{ij}(X_t) + [E_t]_{ij}, \quad t \in [T],$$
where $[Y_t]_{ij}$ denotes the coefficient on the $i^{th}$ row and $j^{th}$ column of the matrix $Y_t$. Riesz representation theorem [5] ensures the existence of a unique matrix $M_{ij}^* \in \mathbb{R}^{m \times q}$ such that

$$[Y_t]_{ij} = \mathrm{Tr}\left(X_t^\top M_{ij}^*\right) + [E_t]_{ij}, \quad t \in [T],$$

where $X_t^\top$ stands for the transposed of $X_t$ and $\mathrm{Tr}\left(X_t^\top M_{ij}^*\right)$ denotes the trace of the square matrix $X_t^\top M_{ij}^*$. In this model, the objective is to estimate the $np$ matrices $(M_{ij}^*)$. This problem is equivalent to considering $np$ independent trace regression models [38]. Hence, this naive approach ignores the multivariate nature of the possibly correlated entries of each response $Y_t$.

To overcome this issue, we consider for $i \in [n]$ and $j \in [p]$ the vectors $\alpha_i \in \mathbb{R}_+^m$ and $\beta_j \in \mathbb{R}^q$ and assume $M_{ij}^* = \alpha_i \beta_j^\top$. For identifiability issues we assume that for all $i \in [n]$, $\|\alpha_i\|_1 = 1$. This model ensures that the matrices $(M_{ij}^*)$ share a common structure. It also implies that they share the same rank, namely one here. Hence this model now accounts for the multivariate nature of the problem. It has $n(m-1) + pq$ free parameters and rewrites as follows:

$$[Y_t]_{ij} = \mathrm{Tr}\left(X_t^\top \alpha_i \beta_j^\top\right) + [E_t]_{ij}, \quad t \in [T].$$

Consider $A^*$ the matrix obtained by stacking the $n$ row vectors $\alpha_i$ and $B^*$ the matrix obtained by stacking the $p$ column vectors $\beta_j$. This leads to the BMLR model (1).

## 1.2 Related works

The BMLR model, first introduced in [23], has gained notable attention as a powerful framework for examining relationships between matrix-structured responses and predictors. In the development of estimation techniques for this model, two principal methods have been explored: alternating minimization and spectral aggregation.

**Alternating Minimization**, usually presented in a Maximum Likelihood Estimation [16] or a least squares Estimation(LSE) context [23]. As the objective is non convex in both parameters, a two-step iterative algorithm is usually derived to construct the estimators.

**Spectral Methods**, presented in a factor model framework in [7], offers an alternative that leverages the spectral properties of the target matrices. Authors propose a new estimation method, the $\alpha$-PCA, that aggregates the information in both first and second moments and extract it via a spectral method. They show that for specific values of the hyperparameter $\alpha$, namely $\alpha = -1$, the $\alpha$-PCA method corresponds to the least squares estimator. However, the procedure is non convex and they rely on approximate solution by alternating minimization. More specifically they maximize row and column variances respectively after projection.

**Kronecker Product Factorization:** The Kronecker Product Factorization (KRO-PRO-FAC) method has recently introduced new possibilities for estimation within matrix-valued linear regression. [11] present this approach, which leverages Kronecker products to decompose complex matrices into simpler components. A key advantage of this method is that it circumvents the need to estimate the covariance between individual entries of the response matrix, significantly reducing computational complexity. The KRO-PRO-FAC algorithm is accompanied by non-asymptotic bounds on the estimation of the Kronecker product between the parameters. However, an important limitation of this approach is the lack of direct control over the estimation accuracy of each parameter separately. This restricts its applicability in scenarios where parameter-wise interpretability or precision is critical, leaving a gap that motivates alternative methodologies, such as the optimization-free estimators proposed in this work.

**Autoregressive Frameworks:** In autoregressive settings, where $X_t := Y_{t-1}$ in (1), the primary focus is on capturing temporal dependencies by minimizing the residual sum of squares [9, 48, 49, 24, 43]. These frameworks are particularly relevant for modeling dynamic systems and matrix-valued time series. However, parameter estimation in autoregressive frameworks typically relies on computationally intensive procedures, such as iterative optimization or matrix decompositions. These methods become increasingly impractical as the dimensions of the data grow, creating a significant bottleneck in high-dimensional settings. This challenge is especially pronounced in applied studies, where the scale and complexity of datasets continue to expand, underscoring the need for more efficient and scalable estimation techniques.

## 1.3 Summary of Contributions

This work studies the estimation problem of the matrix parameters $A^*$ and $B^*$ in (1). Our contributions are the following:

**Noiseless Case Analysis:** In Section 2, we study an oracle case and establish that, in the absence of noise, the true parameters can be exactly retrieved. This analysis highlights the fundamental identifiability properties of the model.

**Optimization-Free Estimators:** In Section 3, we propose explicit, optimization-free estimators $\hat{B}$ and $\hat{A}$ defined in (3) and (4) respectively. They significantly simplify the estimation process and are particularly advantageous in high-dimensional settings where traditional optimization-based methods become computationally prohibitive. Theoretical guarantees are provided in Theorems 3.3 and 3.6. We establish non-asymptotic bounds characterizing the dependence of estimation accuracy on the problem dimensions $(n, p, m, q)$ and the sample size $T$. For $\hat{A} \in \mathbb{R}^{n \times m}$, the performance improves with larger sample size $T$ and larger values of $p$ and $q$, the column dimensions of the response and predictor matrices, respectively. However, the performance deteriorates as $n$ and $m$, which determine the size of $A^*$, increase. In contrast, for $\hat{B} \in \mathbb{R}^{q \times p}$, the convergence rate improves with increases in $T$ and $n$, showcasing a "blessing of dimensionality" effect in the row dimension of the target matrices. Nonetheless, the performance decreases with larger values of $p$, $q$, and $m$.

**Numerical Validation:** In Section 4, we validate our theoretical findings through extensive numerical simulations. On synthetic datasets, we show that the empirical convergence rates align closely with theoretical predictions, see Figure 1. For real-world data from the CIFAR-10 dataset, we demonstrate the practical effectiveness of our procedures by introducing controlled noise, estimating the correction matrices $\hat{A}$ and $\hat{B}$ on the training set, and evaluating denoising performance on the test set. The results, presented in Figures 2 and 3, highlight the effectiveness of the proposed estimators in practical scenarios.

**Sparse Adaptive Estimators:** Extending the framework to sparse settings, we introduce in Appendix 3.3 hard-thresholded estimators, $\hat{B}^S$ and $\hat{A}^S$, defined in (5) and (6) respectively. These estimators exploit the sparsity of the true parameters to achieve improved convergence rates, as established in Theorems 3.7 and 3.8. Moreover, we demonstrate that these estimators can recover the exact support of the true parameters with high probability, providing strong practical guarantees. These estimators exhibit the same dependency on the problem dimensions $(n, p, m, q)$ and the sample size $T$ as their dense counterpart $\hat{B}$ and $\hat{A}$.

## 2 Analysis at the population level

In this oracle case we consider $T$ observations $(X_t, M_t)$ that satisfy the relationship

$$M_t = A^* X_t B^*, \quad t \in [T], \tag{2}$$

where $A^* \in \mathbb{R}_+^{n \times m}$ and $B^* \in \mathbb{R}^{q \times p}$ are the unknown parameters to be recovered. The matrix $A^*$ is assumed to have rows with unit $L_1$-norm, ensuring identifiability of the model.

Understanding this oracle case will allow to establish baseline results that will guide the analysis at the sample level (1) involving noise. The primary goal here is to derive conditions under which the matrices $A^*$ and $B^*$ can be uniquely recovered, as well as to propose efficient algorithms for their recovery. This involves leveraging the structure of the observed predictors $(X_t)_{t=1}^T$ and the constraints on the parameters.

In this section, we assume that the matrices $(X_t)_{t=1}^T$ form a generating family of $\mathbb{R}^{m \times q}$, which implies that $T$ is larger than $mq$. We define two matrices, $\mathbb{M} := \left( \text{vec}(M_1), \ldots, \text{vec}(M_T) \right)^\top \in \mathbb{R}^{T \times np}$ and $\mathbb{X} := \left( \text{vec}(X_1), \ldots, \text{vec}(X_T) \right)^\top \in \mathbb{R}^{T \times mq}$.

*Remark* 2.1. When the design matrices $(X_t)_{t=1}^T$ are generated randomly, $\mathbb{X}^\top \mathbb{X}$ is invertible under mild conditions [13, 36, 39, 40].

We note $E_{(k,l)}$ the canonical basis matrix with 1 at entry $(k, l)$ and define the unobserved matrix $\mathbb{C} \in \mathbb{R}^{mq \times np}$ as $\mathbb{C} := \left( \text{vec}(A^* E_{(k,l)} B^*) \right)_{k \in [m], \, l \in [q]}$, where each row of $\mathbb{C}$ corresponds to the vectorized form of $A^* E_{(k,l)} B^*$. The entry of $\mathbb{C}$ located at the $k + (l-1)m$-th row and $i + (j-1)n$-th column is denoted by $[\mathbb{C}]_{(k,l)}^{(i,j)}$. By construction, each entry of the matrix $\mathbb{C}$ is defined as $[\mathbb{C}]_{(k,l)}^{(i,j)} = [A^*]_{ik} \cdot [B^*]_{lj}$, for all $i \in [n]$, $k \in [m]$, $l \in [q]$, and $j \in [p]$. Moreover, in the model (2), the matrix $\mathbb{C}$ can be exactly reconstructed from the observations, as shown in Lemma C.2 in the supplementary material. Corollary 2.4 shows how $A^*$ and $B^*$ and can be exactly recovered from this quantity

**Proposition 2.2.** *In the model* (2)*, where the design matrices $(X_t)_{t=1}^T$ form a generating family of $\mathbb{R}^{m \times q}$, the parameter matrices $A^* \in \mathbb{R}^{n \times m}$ and $B^* \in \mathbb{R}^{q \times p}$ satisfy the following relationships:*

$$\forall (l, j) \in [q] \times [p] : [B^*]_{lj} = \sum_{k=1}^m [\mathbb{C}]_{(k,l)}^{(i,j)}, \ \forall i \in [n],$$

$$\forall (i, k) \in [n] \times [m] : \quad [A^*]_{ik} = \frac{[\mathbb{C}]_{(k,l)}^{(i,j)}}{[B^*]_{lj}}, \quad \text{for any } (l, j) \in [q] \times [p] \text{ such that } [B^*]_{lj} \neq 0.$$

*Remark* 2.3. For fixed $(l, j) \in [q] \times [p]$, the entries $\left( [\mathbb{C}]_{(k,l)}^{(i,j)} \right)_{(i,k)}$ share the same sign, as each entry is the product of $[A^*]_{ik}$ which is non-negative and $[B^*]_{lj}$.

The following corollary provides a representation of the entries of $A^*$ and $B^*$ as averages. This characterization will be particularly useful at the sample level, leading to plug-in estimators.

**Corollary 2.4.** *Let $\mathcal{D}_0 \subset [p] \times [q]$ denote the set of indices $(j,l)$ such that $[B^*]_{lj} \neq 0$ and let $\mathcal{F} := [n] \times [m]$. Then the entries of the matrices $A^*$ and $B^*$ can be expressed as:*

$$[A^*]_{ik} = \frac{\sum\limits_{(j,l) \in \mathcal{D}_0} [\mathbb{C}]_{(k,l)}^{(i,j)}}{\sum\limits_{(j,l) \in \mathcal{D}_0} [B^*]_{lj}}, \quad [B^*]_{lj} = \frac{1}{n} \sum\limits_{(i,k) \in \mathcal{F}} [\mathbb{C}]_{(k,l)}^{(i,j)}.$$

*Remark* 2.5. The magnitude of $[B^*]_{lj}$ can be expressed as $|[B^*]_{lj}| = \frac{1}{n} \sum\limits_{i=1}^{n} \sum\limits_{k=1}^{n} \left| [\mathbb{C}]_{(k,l)}^{(i,j)} \right|$.

## 3 Analysis at the sample level

We now consider $T$ observations $(X_t, Y_t)$ that satisfy (1). Our objective is to estimate $A^*$ and $B^*$.

From the observations $(X_t, Y_t)_{t=1}^{T}$, we construct $\mathbb{Y} := (\mathrm{vec}(Y_1), \ldots, \mathrm{vec}(Y_T))^\top \in \mathbb{R}^{T \times np}$ and $\mathbb{X} := (\mathrm{vec}(X_1), \ldots, \mathrm{vec}(X_T))^\top \in \mathbb{R}^{T \times mq}$.

We assume that the design matrices $(X_t)_{t=1}^{T}$ form a generating family of $\mathbb{R}^{m \times q}$. This assumption implies that $T \geq mq$ and ensures that $\mathbb{X}^\top \mathbb{X} \in \mathbb{R}^{mq \times mq}$ is invertible. Following Remark 2.1, this is a mild assumption. We further define the unobserved noise matrix $\mathbb{E} := (\mathrm{vec}(E_1), \ldots, \mathrm{vec}(E_T))^\top \in \mathbb{R}^{T \times np}$ and the unobserved signal matrix $\mathbb{M} := (\mathrm{vec}(M_1), \ldots, \mathrm{vec}(M_T))^\top \in \mathbb{R}^{T \times np}$ where $(E_t)_{t=1}^{T}$ are the noise matrices defined in (1) and for $t \in [T]$, $M_t := A^* X_t B^*$.

Following Lemma C.2 we define the unobserved matrix $\mathbb{C} := (\mathbb{X}^\top \mathbb{X})^{-1} \mathbb{X}^\top \mathbb{M} \in \mathbb{R}^{mq \times np}$. To analyze the influence of noise in the estimation process, we define $\mathbb{D} := (\mathbb{X}^\top \mathbb{X})^{-1} \mathbb{X}^\top \mathbb{E} \in \mathbb{R}^{mq \times np}$. Finally, we derive from the observations $\widehat{\mathbb{C}} := (\mathbb{X}^\top \mathbb{X})^{-1} \mathbb{X}^\top \mathbb{Y} \in \mathbb{R}^{mq \times np}$. This leads to $\widehat{\mathbb{C}} = \mathbb{C} + \mathbb{D}$.

We assume that for all $(i,j,k) \in [n] \times [p] \times [q]$, the row sums $\sum_{k=1}^{m} \left[ \widehat{\mathbb{C}} \right]_{(k,l)}^{(i,j)}$ are nonzero. It ensures that the plug-in estimator for $A^*$ is well-defined. Notably, this assumption holds almost surely when the noise matrices $(E_t)_{t=1}^{T}$ are drawn from a continuous distribution, as is the case in this study.

### 3.1 Definition of the estimators

Leveraging the results from Corollary 2.4, we define the plug-in estimators $\hat{B} \in \mathbb{R}^{q \times p}$ of $B^*$, defined for all $(j,l) \in \mathcal{D} := [p] \times [q]$ and the plug-in estimator $\tilde{A} \in \mathbb{R}^{n \times m}$ of $A^*$, defined for all $(i,k) \in [n] \times [m]$, as follows:

$$[\hat{B}]_{lj} := \frac{1}{n} \sum_{(i,k) \in \mathcal{F}} \left[ \widehat{\mathbb{C}} \right]_{(k,l)}^{(i,j)}, \quad [\tilde{A}]_{ik} := \frac{\sum\limits_{(j,l) \in \mathcal{D}} \left[ \widehat{\mathbb{C}} \right]_{(k,l)}^{(i,j)}}{\sum\limits_{(j,l) \in \mathcal{D}} [\tilde{B}]_{lj}^{(i)}}, \tag{3}$$

where $[\tilde{B}]_{lj}^{(i)} := \frac{1}{n-1} \sum\limits_{\substack{r=1 \\ r \neq i}}^{n} \sum\limits_{s=1}^{m} \left[ \widehat{\mathbb{C}} \right]_{(s,l)}^{(r,j)}$. In the definition of $[\tilde{A}]_{ik}$, we ensure that the terms in the numerator do not appear in the denominator to preserve statistical independence of both terms. It is important to note that the entries of $\tilde{A}$ are defined as the ratio of random variables. While the behavior of such ratios has been studied in the literature, particularly in Gaussian cases [17, 34], the results obtained through this approach would require heavy assumptions and remain challenging to interpret in our context.

When the Gaussian variables in the ratio are centered, the distribution of the ratio is known as the Cauchy distribution [25]. However, for non-centered Gaussian variables, the probability density function of the ratio takes a significantly more complex form [22], making the analysis cumbersome. Under certain conditions, it is possible to approximate the ratio with a normal distribution [15], but this requires additional assumptions on the model and would still yield results that are difficult to decipher. Consequently, we opt for a different approach that avoids these complications while retaining interpretability. Specifically, we observe that the plug-in estimator $\tilde{A}$ does not fully exploit

a key property of the model: the entries of the matrix $A^*$ are constrained to lie between $0$ and $1$. This additional structure could be leveraged to improve the estimator's performance. Hence we define the estimator $\hat{A} \in \mathbb{R}^{n \times m}$ of $A^*$, defined for all $(i,k) \in [n] \times [m]$, as:

$$[\hat{A}]_{ik} := \max\left(0, \ \min\left([\tilde{A}]_{ik}, 1\right)\right). \tag{4}$$

## 3.2 Theoretical analysis with known variance

We present the matrix normal case with known fixed variance under the ORT assumption. The matrix normal distribution generalizes the multivariate normal distribution to matrix-valued random variables [20, 1, 35]. The sparse case is presented in the section 3.3.

**Lemma 3.1.** *Under the assumption that $\mathbb{X}^\top \mathbb{X}$ is full rank, the matrix $\widehat{\mathbb{C}} := \left(\mathbb{X}^\top \mathbb{X}\right)^{-1} \mathbb{X}^\top \mathbb{Y} \in \mathbb{R}^{mq \times np}$ satisfies $\widehat{\mathbb{C}} \sim \mathcal{MN}_{mq \times np}\left(\mathbb{C}, \left(\mathbb{X}^\top \mathbb{X}\right)^{-1}, \sigma^2 I_{np}\right).$*

Following the vast literature on linear regression and Gaussian sequence models [44, 19, 37, 12], we make the ORT assumption to capture a better understanding of the phenomenon at play. This assumption serves primarily to facilitate the theoretical analysis. Notably, the numerical experiments in Section 4 are conducted without relying on the ORT condition. We discuss relaxation of the ORT assumption in Appendix A.

**Assumption 3.2** (ORT assumption). We assume that the design matrix $\mathbb{X}$ satisfies $\mathbb{X}^\top \mathbb{X} = T \cdot I_{mq}$.

Under Assumption 3.2, Lemma 3.1 ensures that the entries of $\widehat{\mathbb{C}}$ are independent and normally distributed. The following theorem establishes non-asymptotic upper bounds on the convergence rates of $\hat{B}$ under various norms.

**Theorem 3.3.** *Under Assumption 3.2, the estimator $\hat{B}$ introduced in (3) satisfies the following non-asymptotic bounds for any $\epsilon > 0$:*

$$\mathbb{P}\left[\|\hat{B} - B^*\|_+ > \epsilon\right] \leq \frac{pq\sigma\sqrt{2m}}{\epsilon\sqrt{Tn\pi}} \exp\left(-\frac{Tn\epsilon^2}{2m\sigma^2}\right), \ \mathbb{P}\left[\|\hat{B} - B^*\|_F^2 > \epsilon\right] \leq 2\exp\left(-\frac{Tn\epsilon}{3m\sigma^2} + \frac{2pq}{3}\right),$$

$$\mathbb{P}\left[\|\hat{B} - B^*\|_{\mathrm{op}} > \epsilon\right] \leq 2\exp\left(-\left(\frac{\epsilon\sqrt{nT}}{2\sigma\sqrt{m}} - \psi_{p,q}\right)^2\right).$$

*Here, $\psi_{p,q} := \dfrac{\sqrt{p} + \sqrt{q}}{2}$, $\|\cdot\|_+$ is the elementwise maximum norm, $\|\cdot\|_{\mathrm{op}}$ is the operator norm, and $\|\cdot\|_F$ is the Frobenius norm.*

We observe that the convergence rate of the estimator $\hat{B}$ exhibits the anticipated dependence on the sample size $T$, improving as the number of observations increases. Notably, our analysis reveals a "blessing of dimensionality" effect, wherein the convergence rate accelerates as the row dimension $n$ of the observed target matrices $(Y_t)_{t=1}^T$ grows. Conversely, the convergence rate is negatively affected by the size $pq$ of $B^*$. Furthermore, the variance parameter $\sigma$ also exerts a detrimental influence on the convergence rate, as intuitively expected.

The following Lemma provides a probabilistic control over the event where the plug-in estimator $\tilde{A}$, defined in (3), coincides with its modified version $\hat{A}$ defined in (4). We assume that the entries of $A^*$ are strictly bounded from below by $0$ and from above by $1$. We also assume that the sum of the entries of $B^*$ is positive.

**Assumption 3.4.** We assume that for all $(i,k) \in [n] \times [m]$ we have $0 < [A^*]_{ik} < 1$. In addition we assume that $\beta^* > 0$ where $\beta^* := \dfrac{1}{pq} \sum_{j=1}^p \sum_{l=1}^q [B^*]_{lj}$.

In model (1), $A^*$ has non-negative entries with rows summing to one, so its entries lie in $[0,1]$. Assumption 3.4 strengthens this to entries in $]0,1[$. The sparse case is discussed in Appendix 3.3.

**Lemma 3.5.** *Under Assumptions 3.2 and 3.4, the estimators $\tilde{A}$ and $\hat{A}$ introduced in (3) and (4) satisfy the following property for any $(i,k) \in [n] \times [m]$:*

$$\mathbb{P}\left[[\tilde{A}]_{ik} \neq [\hat{A}]_{ik}\right] \leq \frac{\tilde{\sigma}}{\mu_{ik}\sqrt{2pqT\pi}} \exp\left(-\frac{Tpq\mu_{ik}^2}{2\tilde{\sigma}^2}\right) + \frac{\sigma}{\nu_{ik}\sqrt{2pqT\pi}} \exp\left(-\frac{Tpq\nu_{ik}^2}{2\sigma^2}\right),$$

*where $\mu_{ik} := \frac{1}{pq} \sum_{l=1}^{q} \sum_{j=1}^{p} [B^*]_{lj} (1 - [A^*]_{ik})$, $\nu_{ik} := [A^*]_{ik}\beta^*$ and $\tilde{\sigma} := \sigma\sqrt{1 + \frac{m}{n-1}}$.*

We first note that under Assumption 3.4, the quantities $\mu_{ik}$ and $\nu_{ik}$ are positive. We then observe that as the sample size $T$ increases, the plug-in estimator $\tilde{A}$ is more likely to coincide with its modified version $\hat{A}$. The intuition behind this result is straightforward: as the sample size grows, $[\tilde{A}]_{ik}$ converges to $[A^*]_{ik}$, which inherently lies between 0 and 1. Consequently, the modifications introduced in $\hat{A}$ become unnecessary as $[\tilde{A}]_{ik}$ naturally satisfies the constraints of the model. A similar phenomenon occurs as $p$, the number of columns of the response matrices, and $q$, the number of columns of the predictor matrices, increase. Larger $p$ and $q$ effectively provide more information about the structure of the model, leading to improved accuracy of the plug-in estimator $\tilde{A}$ and reducing the need for corrections by $\hat{A}$.

The following theorem provides a finite-sample analysis of the performance of $\hat{A}$.

**Theorem 3.6.** *Under Assumptions 3.2 and 3.4, the estimator $\hat{A}$ introduced in* (4) *satisfies for any $\epsilon > 0$:*

$$\mathbb{P}\left[\left\|\hat{A} - A^*\right\|_+ > \frac{2\epsilon}{|\beta^*|}\right] \leq \frac{nm\sigma\sqrt{2}}{\epsilon\sqrt{pqT\pi}} \left[\exp\left(-\frac{Tpq\epsilon^2}{2\sigma^2}\right) + \frac{\sqrt{m}}{\sqrt{(n-1)}} \exp\left(-\frac{Tpq(n-1)\epsilon^2}{2m\sigma^2}\right)\right]$$

$$+ \frac{nm}{\sqrt{2pqT\pi}} \left[\frac{\tilde{\sigma}\exp\left(-\frac{Tpq\mu^2}{2\tilde{\sigma}^2}\right)}{\mu} + \frac{\sigma\exp\left(-\frac{Tpq\nu^2}{2\sigma^2}\right)}{\nu}\right],$$

*where $\nu := \min_{i,k} \nu_{ik}$, $\mu := \min_{i,k} \mu_{ik}$ with $\mu_{ik}$, $\nu_{ik}$ and $\tilde{\sigma}$ defined in Lemma 3.5.*

We observe that the finite-sample performance of the estimator $\hat{A}$ improves with the sample size $T$, reflecting the benefit of more observations. Additionally, the convergence rate is positively influenced by increases in the column dimension $p$ of the observed target matrices $(Y_t)_{t=1}^{T}$ and the column dimension $q$ of the predictor matrices $(X_t)_{t=1}^{T}$. This reflects a "blessing of dimensionality" effect, as additional columns provide richer information for the estimation process. Notably, this behavior contrasts with that of $\hat{B}$, as detailed in Theorem 3.3, where increases in $p$ and $q$ have a detrimental impact. Moreover, the magnitude of $\beta^*$ plays a crucial role in determining the convergence rate, with larger values of $|\beta^*|$ leading to faster convergence. Conversely, the performance of $\hat{A}$ deteriorates as the size $nm$ of $A^*$ increases and the variance parameter $\sigma$ negatively affects the convergence rate. Higher noise levels degrade the accuracy. We note that the degradation of both rates with increasing $m$ breaks the symmetry between $A^*$ and $B^*$. This is because of the assumption we impose on $A^*$.

## 3.3 Sparse Case with Known Variance

In this part, we extend our theoretical analysis to incorporate sparsity assumptions on the parameters $A^*$ and $B^*$. By leveraging the sparse structure, we aim to develop estimation strategies tailored for high-dimensional settings, where many entries of the parameters are expected to be zero. This sparse framework addresses practical scenarios where dimensionality reduction is critical.

We propose the hard-thresholding estimator [19, 37] $\hat{B}^S$, defined as follows for all $(l, j) \in [q] \times [p]$:

$$[\hat{B}^S]_{lj} = [\hat{B}]_{lj} \cdot \mathbf{1}_{\{|[\hat{B}]_{lj}| > 2\tau\}} \tag{5}$$

where $[\hat{B}]_{lj}$ are the entries of the initial estimator $\hat{B}$ defined in (3), and $\tau > 0$ is a user-defined threshold. The hard-thresholding operation enforces sparsity by setting small entries of $\hat{B}$ to zero, aligning the estimator with the sparse structure of $B^*$.

The following theorem establishes a non-asymptotic upper bound on the convergence rate of $\hat{B}^S$ for the Frobenius norm under sparsity assumptions on the parameter $B^*$.

**Theorem 3.7.** *Under Assumption 3.2, for any $\delta \in (0, 1)$, the estimator $\hat{B}^S$ introduced in* (5), *with the threshold $\tau := \sigma\sqrt{\frac{2m}{nT}}\left(\sqrt{\log(2pq)} + \sqrt{\log\left(\frac{2}{\delta}\right)}\right)$, enjoys the following non-asymptotic properties on the same event holding with probability at least $1 - \delta$:*

1. *If $\|B^*\|_0$ denotes the number of nonzero coefficients in $B^*$, then*

$$\|\hat{B}^S - B^*\|_F^2 < 16\,\|B^*\|_0\,\tau^2.$$

2. *If the entries of $B^*$ satisfy $\min_{l\in[q],j\in[p]}|[B^*]_{lj}| > 3\tau$, then the support of $\hat{B}^S$ perfectly matches that of $B^*$, namely $\text{supp}(\hat{B}^S) = \text{supp}(B^*)$, where $\text{supp}(\cdot)$ denotes the set of indices corresponding to the nonzero entries of a matrix.*

Theorem 3.7 highlights the performance of the sparse estimator $\hat{B}^S$ under sparsity assumptions on the true parameter $B^*$. The convergence rate of $\hat{B}^S$ improves as the sparsity of $B^*$ increases, demonstrating the benefits of leveraging sparse structures. The threshold $\tau$ exhibits favorable scaling with the sample size $T$ and row dimension $n$, both of which contribute to reducing $\tau$, enhancing the estimator's performance. Conversely, $\tau$ increases with the dimensions $p$ and $q$, reflecting the greater challenge of estimation in higher-dimensional settings. Moreover the threshold $\tau$ scales with $\sigma$, capturing the adverse impact of higher noise levels on the estimation accuracy. This emphasizes the observations from Theorem 3.3. Finally the condition on the minimum magnitude of the entries of $B^*$ ensures that the threshold $\tau$ enables exact recovery of the support of $B^*$, with high probability.

We now propose the hard-thresholding estimator $\hat{A}^S$, defined as follows for all $(i,k) \in [n] \times [m]$ where $\hat{\gamma}_{ik} := \frac{1}{pq} \sum_{j=1}^{p} \sum_{l=1}^{q} \left[\widehat{\mathbb{C}}\right]_{(k,l)}^{(i,j)}$:

$$[\hat{A}^S]_{ik} = [\hat{A}]_{ik} \cdot \mathbf{1}_{\{|\hat{\gamma}_{ik}|>2\tau\}} \tag{6}$$

The following theorem establishes a non-asymptotic upper bound on the convergence rate of $\hat{A}^S$ for the Frobenius norm under sparsity assumptions on the parameter $A^*$.

**Theorem 3.8.** *Under Assumption 3.2, for any $\delta \in (0,1)$, the estimator $\hat{A}^S$ defined in (6), with the threshold $\tau := \sigma\sqrt{\frac{2}{Tpq}}\left(\sqrt{\log(2nm)} + \sqrt{\log\left(\frac{2}{\delta}\right)}\right)$, satisfies the following non-asymptotic properties on an event holding with probability at least $1 - 2\delta$:*

1. *If $\|A^*\|_0$ denotes the number of nonzero coefficients in $A^*$, then*

$$\|\hat{A}^S - A^*\|_F^2 \le |\beta^*|^{-2}\|A^*\|_0\,(2t_\delta + 3\tau)^2,$$

*where $\beta^* := \frac{1}{pq}\sum_{j=1}^{p}\sum_{l=1}^{q}[B^*]_{lj}$. and $t_\delta$ satisfies $\frac{\sigma\sqrt{2m}}{t_\delta\sqrt{npqT\pi}}\exp\left(-\frac{Tpqt_\delta^2}{2m\sigma^2}\right) + \frac{\sigma\sqrt{2}}{t_\delta\sqrt{pqT\pi}}\exp\left(-\frac{Tpqt_\delta^2}{2\sigma^2}\right) = \delta$.*

2. *If the entries of $A^*$ satisfy $\min_{i\in[n],k\in[m]}|[A^*]_{ik}| > 3\tau$ and if $3\beta^* > 1$, then the support of $\hat{A}^S$ perfectly matches that of $A^*$, namely $\text{supp}(\hat{A}^S) = \text{supp}(A^*)$.*

*Remark* 3.9. The parameter $t_\delta$, which determines the concentration properties of the estimator, decreases as $T$, $p$, and $q$ increase. Consequently, the estimator $\hat{A}^S$ benefits from improved performance as the sample size $T$ and latent dimensions $p$ and $q$ grow.

Theorem 3.8 characterizes the non-asymptotic properties of the sparse estimator $\hat{A}^S$ under sparsity assumptions on the true parameter matrix $A^*$. The first result provides a Frobenius norm error bound that scales with the sparsity level $\|A^*\|_0$. This bound is inversely related to the squared magnitude of $\beta^*$ (the average entry in $B^*$), indicating that stronger signals in the underlying parameter matrix $B^*$ lead to improved estimation accuracy. This is similar to the phenomenon described in Theorem 3.6. The error bound also depends on both the threshold parameter $\tau$ and the concentration parameter $t_\delta$, which capture the impact of noise and sample size on the estimation performance. The threshold $\tau$ exhibits several important dependencies. It decreases with the sample size $T$, reflecting improved estimation with more observations. It similarly decreases with dimensions $p$ and $q$, showcasing a beneficial effect of higher dimensionality. Conversely it scales with the noise level $\sigma$, capturing the detrimental impact of increased noise levels. Finally it grows logarithmically with the matrix dimensions $n$ and $m$, indicating a mild sensitivity to the size of $A^*$. The second result establishes conditions for perfect support recovery of $A^*$. Specifically, when the minimum magnitude of the

nonzero entries in $A^*$ exceeds three times the threshold $\tau$, and the average effect $\beta^*$ is sufficiently large ($3\beta^* > 1$), the sparse estimator exactly recovers the support of $A^*$.

As noted in Remark 3.9, the concentration parameter $t_\delta$ decreases with larger values of $T$, $p$, and $q$. This property, combined with the similar behavior of $\tau$, demonstrates that the sparse estimator benefits from both increased sample size and higher latent dimensions, a particularly favorable characteristic for high-dimensional settings.

## 4 Numerical Simulations

### 4.1 Synthetic data

Now we evaluate the performance of the proposed estimators through numerical simulations.
**Simulation Setup**: The simulations involve the generation of matrices $A^*$, $B^*$, $X_t$, $E_t$ and $Y_t$. By default, the parameters are set as $n = 15$, $m = 13$, $p = 14$, $q = 12$, $T = 2000$, and $\sigma = 1$. These default parameter values are adjusted to analyze the effects of $n$, $m$, $p$, $q$, $T$, and $\sigma$ on the performance of the proposed estimators. $A^*$ is a $n \times m$ matrix with random entries sampled from a uniform distribution over $[0, 1)$ and rows then normalized to sum to 1. $B^*$ is a $q \times p$ matrix with random entries sampled from a uniform distribution over $[0, 1)$. $(X_t)$ is a sequence of $T$ matrices of size $m \times q$ with random entries sampled from a uniform distribution over $[0, 1)$. $(E_t)$ is a sequence of $T$ noise matrices of size $n \times p$, with entries drawn from a Gaussian distribution with mean 0 and standard deviation $\sigma = 1$. $(Y_t)$ is a sequence of $T$ observation matrices, where $Y_t = A^* X_t B^* + E_t$.

**Estimation and Evaluation**: The estimators $\hat{A}$ and $\hat{B}$ are computed using (4) and (3) respectively. To evaluate their performances, we vary the parameters $n$, $m$, $p$, $q$, and $T$ to observe their impact on the estimation accuracy. For each parameter setting, we compute the Frobenius norm of the errors $\|\hat{A} - A\|_F$ and $\|\hat{B} - B\|_F$ together with the Operator norm of the errors $\|\hat{A} - A\|_{op}$ and $\|\hat{B} - B\|_{op}$. These errors are plotted as functions of the varying parameters in Figure 1.

**Validation of Theoretical Properties**: From the plots, we observe that as $T$ increases, Figure 1e, the errors in $\hat{A}$ and $\hat{B}$ decrease, indicating improved estimation accuracy with more data. As $p$ and $q$ increase, Figures 1c and 1d, the errors in $\hat{A}$ decrease and the errors in $\hat{B}$ increase. As $m$ increases, Figures 1b, both the errors in $\hat{A}$ and $\hat{B}$ increase. As $n$ increases, Figure 1a, the errors in $\hat{A}$ increase and the errors in $\hat{B}$ decrease. These results confirm the theoretical properties of the estimators, detailed in Theorems 3.6 and 3.3. Additionally, we have performed numerical simulations to support the statement from Corollary 2.4. Appendix B.2 provides additional experiments.

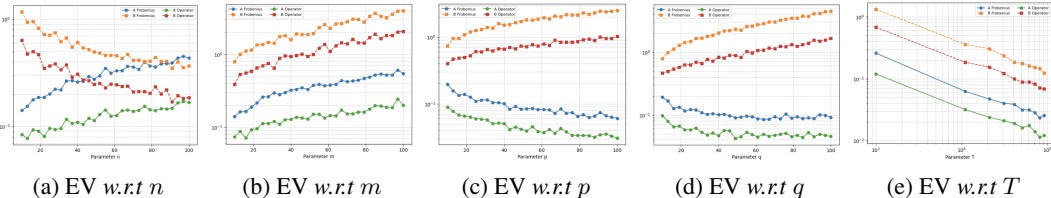

| (a) EV *w.r.t* $n$ | (b) EV *w.r.t* $m$ | (c) EV *w.r.t* $p$ | (d) EV *w.r.t* $q$ | (e) EV *w.r.t* $T$ |

Figure 1: Evolution (EV) of the Frobenius norm (resp. operator norm) of $\hat{A} - A^*$ (in blue, resp. in green) and of $\hat{B} - B^*$ (in orange, resp. in red) with respect to (*w.r.t.*) different parameters.

We observe that the empirical error rates align closely with the theoretical rates (derived under ORT). It suggests that deviations from orthogonality may lead only to a mild degradation in performance. Moreover, the degradation of both rates with increasing $m$ is confirmed by the simulations (Figure 1b). Thus, this observed asymmetry is not an artifact of loose analysis, but a consequence of the model's structural assumptions.

### 4.2 Real-world data

We also evaluate our proposed methods on real-world data using the CIFAR-10 dataset. It contains 50,000 training and 10,000 test RGB images, each of size $32 \times 32 \times 3$. The pixel values are nor-

malized to $[0, 1]$ for computational consistency. Our goal is to simulate noisy image transformations and assess the effectiveness of the correction techniques.

**Noisy Transformations**: We simulate noisy image transformations via left and right matrix multiplications with $A^*$ and $B^*$ respectively. First, we define $A = I_{32} + \epsilon E_1$, where $E_1$ is a $32 \times 32$ matrix with i.i.d. entries from a standard Gaussian distribution, and $A^*$ is obtained by normalizing each row of $A$ to sum to 1. Similarly, matrix $B^*$ is given by $B^* = I_{32} + \epsilon E_2$, where $E_2$ is a $32 \times 32$ matrix with i.i.d. standard Gaussian entries. The parameter $\epsilon$ controls the noise level in both transformations. For both training and test images, the noisy transformation is applied independently to each color channel. The transformation for a given channel $c \in \{1, 2, 3\}$ (corresponding to red, green, and blue) is defined as $X_{\text{noisy}}^{(c)} = (A^*)^{-1} \cdot X_{\text{or}}^{(c)} \cdot (B^*)^{-1}$, where $X_{\text{or}}^{(c)}$ represents the $c^{\text{th}}$ channel of the original image and $X_{\text{noisy}}^{(c)}$ is the corresponding noisy version.

**Correction Process**: The correction process is learnt on the training set. From the noisy transformed images, we estimate the correction matrices $\hat{A}_{\text{train}}^{(c)}$ and $\hat{B}_{\text{train}}^{(c)}$ using (4) and (3), processing color channel independently. Once the correction matrices $\hat{A}_{\text{train}}^{(c)}$ and $\hat{B}_{\text{train}}^{(c)}$ are computed, they are applied to the noisy test images to reconstruct the corrected test images. For each channel in a test image, the corrected channel is computed as $X_{\text{corr}}^{(c)} = \hat{A}_{\text{train}}^{(c)} \cdot X_{\text{noisy}}^{(c)} \cdot \hat{B}_{\text{train}}^{(c)}$. Figure 2 shows an example from the test set, illustrating the original image, its noisy version for $\epsilon = 0.02$, and the corresponding corrected image respectively.

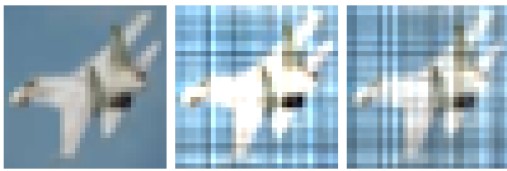

Figure 2: Original, noisy, and corrected versions of the $11^{\text{th}}$ image from the test set for $\epsilon = 0.02$.

**Evaluation of Correction Quality**: To evaluate the effectiveness of the correction process, we compute $D_{\text{o,n}} := \sum_{c=1}^{3} \|X_{\text{or}}^{(c)} - X_{\text{noisy}}^{(c)}\|_F^2$, the Frobenius distance between the original image and its noisy version, and $D_{\text{o,c}} = \sum_{c=1}^{3} \|X_{\text{or}}^{(c)} - X_{\text{corr}}^{(c)}\|_F^2$ the Frobenius distance between the original image and its corrected version. We plot in Figure 3 $D_{\text{o,n}}$ and $D_{\text{o,c}}$ as functions of the noise factor $\epsilon$ averaged over the entire test set. Appendix B.3 presents additional plots showing how reconstruction accuracy varies with the effective signal-to-noise ratio (SNR) under both Frobenius and max norms.

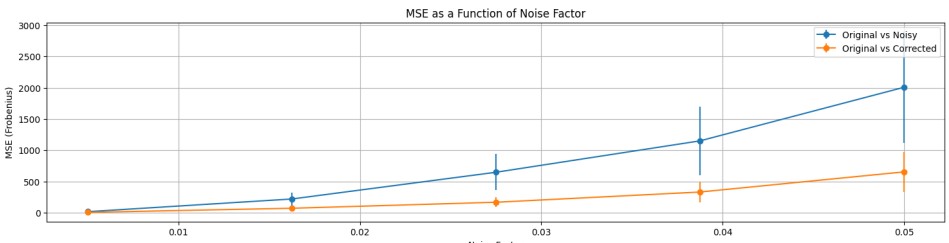

Figure 3: $D_{\text{o,n}}$ (blue) and $D_{\text{o,c}}$ (orange), averaged on the test set, as functions of $\epsilon$. Error bars indicate standard deviations.

**Conclusion**

The results demonstrate that the proposed correction process effectively mitigates the impact of noise. The corrected images closely approximate the original images, as shown by both qualitative (Figure 2) and quantitative (Figure 3) metrics. This methodology generalizes well to real-world data, underscoring the applicability of our framework beyond synthetic simulations.

## Acknowledgments

This research was carried out while the author was affiliated with Cornell University, where the main theoretical development and experiments were conducted. The completion of the manuscript and submission process took place after joining Capital Fund Management (CFM). The author gratefully acknowledges the academic environment at Cornell for fostering this work and the support of CFM during the finalization of the paper.

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

# A  On the Validity and Relaxation of Assumptions

In the main text, we assume that the noise matrices $(E_t)_{t=1}^T$ have independent entries drawn from a centered Gaussian distribution with constant variance $\sigma^2$. This corresponds to a matrix normal distribution with isotropic row and column covariances, and facilitates clean non-asymptotic analysis. However, our results can be extended to more general settings under mild additional technical effort.

In some proofs, we extend the analysis to a more general setting in which each $E_t$ is drawn independently from a matrix normal distribution $\mathcal{MN}_{n \times p}(0, \Sigma_r, \Sigma_c)$ [14, 47, 20, 1, 35], where $\Sigma_r \in \mathbb{R}^{n \times n}$ and $\Sigma_c \in \mathbb{R}^{p \times p}$ capture row-wise and column-wise dependencies in the noise. Imposing the normalization condition $\Sigma_c \otimes \Sigma_r = \sigma^2 I_{np}$ corresponds to the setting presented in the core of the paper. Under this condition, the general matrix normal setting is equivalent to the isotropic case presented in the core paper.

This formulation allows us to present certain proofs, such as Lemma 3.1, within a broader framework.

**ORT assumption.** Consider the setting where each $E_t$ follows a matrix normal distribution $\mathcal{MN}_{n \times p}(0, \Sigma_r, \Sigma_c)$, where $\Sigma_c \otimes \Sigma_r = \sigma^2 I_{np}$ but without the ORT assumption. In this relaxed setting, Lemma 3.1 remains valid and continues to characterize the distribution of the central quantity $\widehat{\mathbb{C}}$, although the matrix $\mathbb{D}$ now exhibits an anisotropic row covariance structure.

Extending the proofs of Theorems 3.6 and 3.3 to this setting is conceptually straightforward, but technically more involved. One would need to rely on concentration bounds for random matrices with independent but non-isotropic columns. These are available, for instance, in Section 5.5 of [46]. In this case, collinearities in $\mathbb{X}^\top \mathbb{X}$ would naturally appear in the resulting bounds, as illustrated in results such as Theorem 5.62 of the same reference

**Homoskedasticity.** Assuming $\Sigma_c \otimes \Sigma_r = \sigma^2 I_{np}$, or equivalently that $(E_t)_{t=1}^T$ have independent entries drawn from a centered Gaussian distribution with constant variance $\sigma^2$ plays a role analogous to the homoskedasticity assumption in classical linear regression. It reflects a setting in which the signal is entirely explained by the model, and the residuals are unstructured. While restrictive, this assumption is standard in theoretical analysis and provides a tractable foundation for deriving error bounds.

Relaxing this assumption to allow for general row-wise dependence while keeping column independence, still under the ORT Assumption, would require concentration results for random matrices with independent but non-identically distributed rows (see Section 5.4 of [46]). In this case, the noise term in our analysis would be governed by the full tensor-product covariance $\Sigma_c \otimes \Sigma_r$, which would explicitly appear in the resulting bounds (e.g., Theorem 5.44 in [46]).

Relaxing both assumptions simultaneously presents a more significant challenge. To the best of our knowledge, current probabilistic tools do not yet offer sharp and tractable results in this fully general setting. However, this remains a promising direction for future work, potentially requiring new matrix concentration inequalities tailored to specific structured settings.

In summary, although the core analysis assumes isotropic Gaussian noise for clarity and tractability, the main techniques extend to more general noise structures under appropriate assumptions and with access to suitable matrix concentration inequalities.

# B  Additional Numerical Analyses

This appendix presents complementary numerical studies that assess the robustness of the proposed estimators and quantify the alignment between empirical convergence rates and the theoretical predictions.

## B.1  Robustness to the Distribution and Normalization of $A^*$ and $B^*$

To verify that the simulation results in Section 4 are not sensitive to the amplitude or normalization of the true parameters, we repeated all experiments with entries of both $A^*$ and $B^*$ independently

drawn from a $\mathrm{Uniform}[0, c)$ distribution with $c \in \{1, 2, 3, 5\}$, followed by row-wise $\ell_1$ normalization of $A^*$ to ensure identifiability. Across all values of $c$, the empirical convergence rates and qualitative dependencies with respect to the parameters $(n, m, p, q, T)$ remain unchanged. This confirms that the results reported in Figure 1 are not specific to the original choice $c = 1$.

Theoretical analysis shows that the model is identifiable once one of the parameter matrices is normalized. If instead the normalization were imposed on $B^*$, the algebraic expressions of Proposition 2.2 and Corollary 2.4 would be modified by exchanging the roles of $m$ (the number of columns of $A^*$) and $q$ (the number of rows of $B^*$). Consequently, the dependencies on these dimensions in the non-asymptotic bounds of Theorems 3.3 and 3.6 would also be interchanged. This structural asymmetry stems from the identifiability constraint itself and highlights that several normalization choices are possible. The decision to normalize $A^*$ is primarily motivated by interpretability—its nonnegative, row-stochastic structure aligns with the notion of activation or mixing weights—and by analytical tractability of the resulting expressions.

## B.2 Quantitative Comparison Between Empirical and Theoretical Rates

This section reports the quantitative comparison between the empirical convergence slopes of the estimators $(\hat{A}, \hat{B})$ and the theoretical predictions derived from the finite-sample analysis. The objective is to evaluate how the estimation error scales with each model dimension $(n, m, p, q)$ and with the sample size $T$ under three norms: Frobenius, operator, and maximum absolute.

**Experimental setup.** For each parameter $d \in \{n, m, p, q, T\}$, we generated independent datasets while keeping all other quantities fixed, recomputed $(\hat{A}, \hat{B})$, and measured their reconstruction errors

$$\mathrm{Err}_\square(d) = \|\hat{M} - M^*\|_\square, \qquad \square \in \{\| \cdot \|_\mathrm{F}, \| \cdot \|_\mathrm{op}, \| \cdot \|_\mathrm{max}\}.$$

The dependence of $\log(\mathrm{Err}_\square(d))$ on $d$ was then fitted with a linear model

$$\log(\mathrm{Err}_\square(d)) = \alpha_{\square, d} + s_d^{(\square)} f(d) + \varepsilon,$$

where the function $f(d)$ corresponds to:

$$f(d) = \begin{cases} \log d, & d = T \quad \text{(log–log regression)}, \\ d, & d \in \{n, m, p, q\} \quad \text{(linear–log regression)}. \end{cases}$$

This distinction reflects that, in the simulations, both axes were represented on logarithmic scale for $T$, while for $(n, m, p, q)$ the $x$-axis was linear and only the $y$-axis (error) was plotted in log scale. The fitted slope $s_d^{(\square)}$ quantifies the empirical rate of variation of the estimation error with respect to $d$. For the sample size $T$, the theoretical prediction is $s_T^{(\square)} = -\frac{1}{2}$.

**Empirical slopes.** Table 1 summarizes the fitted slopes for all parameters and norms. Positive slopes indicate that the error increases with the corresponding dimension, while negative slopes indicate a decrease.

Table 1: Empirical slopes $s_d^{(\square)}$ of $\log(\mathrm{Err})$ vs. $\log(d)$ for $\hat{A}$ and $\hat{B}$ under the three norms.

| Parameter | $\hat{A}$ | | | $\hat{B}$ | | |
|---|---|---|---|---|---|---|
| | Max | Frobenius | Operator | Max | Frobenius | Operator |
| $n$ | +0.003 | +0.011 | +0.008 | −0.011 | −0.011 | −0.012 |
| $m$ | +0.007 | +0.015 | +0.011 | +0.018 | +0.016 | +0.016 |
| $p$ | −0.012 | −0.011 | −0.011 | +0.002 | +0.011 | +0.008 |
| $q$ | −0.005 | −0.005 | −0.005 | +0.008 | +0.017 | +0.013 |
| $T$ | −0.532 | −0.496 | −0.474 | −0.579 | −0.526 | −0.526 |

**Findings.** Several patterns emerge consistently across all norms:

- **Sample-size dependence.** For both $\hat{A}$ and $\hat{B}$, the slopes with respect to $T$ lie between $-0.58$ and $-0.47$, matching the theoretical prediction $s_T = -\frac{1}{2}$. This confirms that estimation errors decay at the expected $T^{-1/2}$ rate.

- **Dimensional dependencies.** For $\hat{A}$, errors increase with $m$ and decrease with $(p, q)$, whereas for $\hat{B}$ the opposite trend holds—errors increase with $(m, p, q)$ but slightly decrease with $n$. This asymmetric pattern is consistent with the theoretical structure of the model, where the identifiability constraint on $A^*$ leads to mirrored dependencies in $\hat{A}$ and $\hat{B}$.

- **Norm-specific behavior.** The Frobenius norm shows the strongest sensitivity to dimensional changes, reflecting its dependence on all matrix entries; the operator norm shows a weaker dependence consistent with a $(\sqrt{p} + \sqrt{q})$ scaling; and the maximum norm lies between these two regimes.

- **Goodness of fit.** The linear relationships between $\log(\text{Err})$ and $\log(d)$ exhibit high explanatory power for most dimensions, with $R^2$ values above 0.9 in the majority of cases, confirming the robustness of the observed trends.

**Plots.** Figure 4 illustrates the empirical error curves for all parameters. Each subplot reports the average reconstruction error (in logarithmic scale) for $\hat{A}$ and $\hat{B}$ under the three norms. The first four panels correspond to variations in the structural dimensions $(n, m, p, q)$, while the bottom panel shows the dependence on the sample size $T$. The trends confirm the fitted slopes in Table 1: errors decrease approximately linearly on the log scale as $T$ grows, consistent with the $T^{-1/2}$ rate, and vary smoothly with $(n, m, p, q)$ according to the theoretical sign pattern. Notably, $\hat{B}$ exhibits larger sensitivity to $q$ and smaller sensitivity to $n$, while $\hat{A}$ shows the opposite, reflecting the asymmetric normalization of $A^*$. Across all panels, the Frobenius norm produces the steepest gradients, the operator norm the weakest, and the maximum norm lies in between, reproducing the hierarchy predicted by the theoretical bounds.

**Summary.** The empirical analysis confirms that the proposed estimators obey the predicted non-asymptotic scaling laws. The $T^{-1/2}$ convergence rate holds precisely, and the dependence on $(n, m, p, q)$ follows the signs and magnitudes expected from the theoretical bounds. These quantitative findings validate that the theoretical error expressions accurately capture the dominant sources of variation in finite-sample performance.

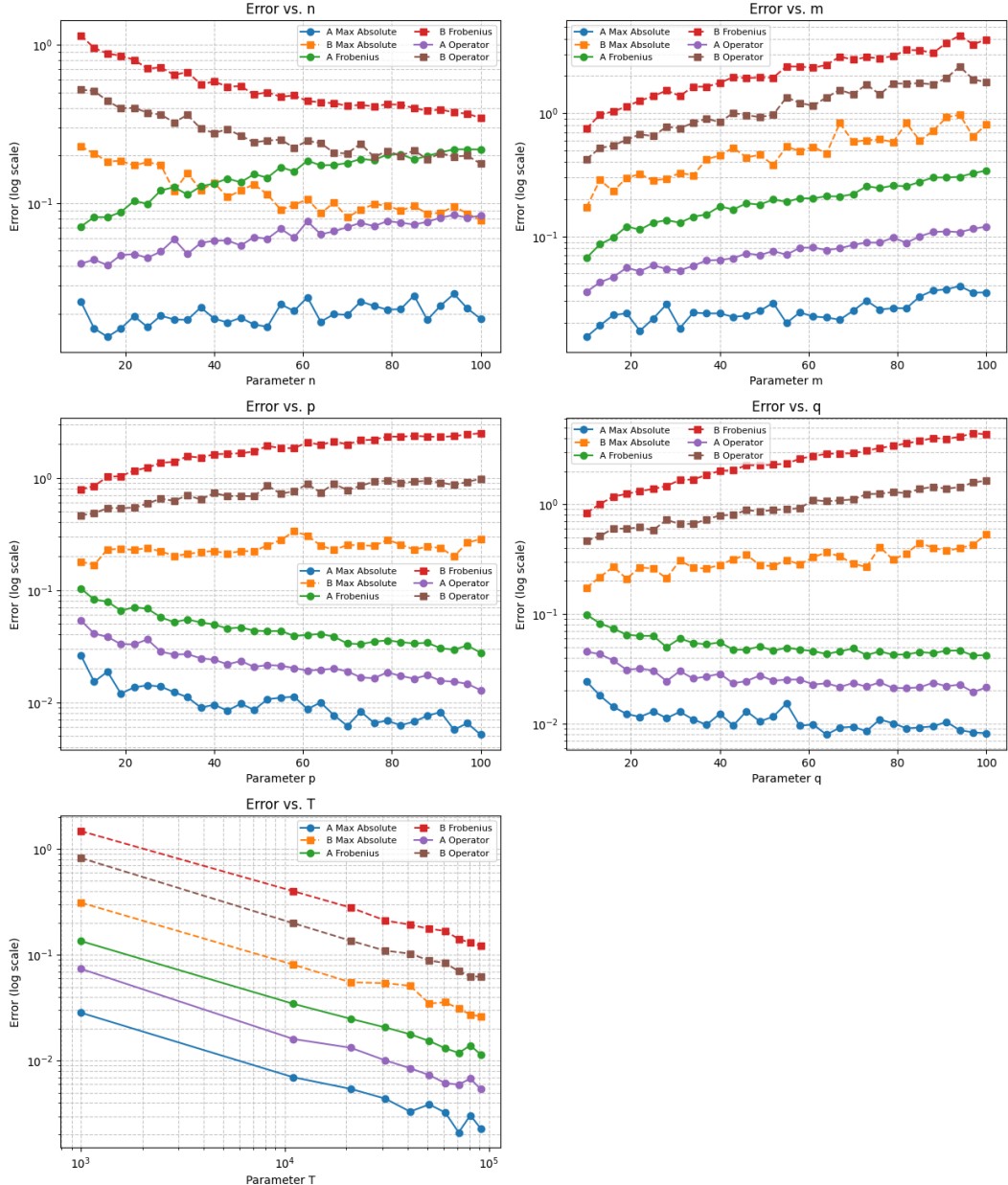

Figure 4: Empirical reconstruction errors of $\hat{A}$ and $\hat{B}$ versus model dimensions and sample size $T$, displayed on a logarithmic scale. Each curve corresponds to a given norm (Frobenius, operator, or max).

## B.3 Reconstruction Error as a Function of Noise Level

To further evaluate robustness to noise, we examined the reconstruction quality of the bilinear estimator under controlled perturbations of $A^*$ and $B^*$. For each noise level $\epsilon$, we generated perturbed matrices

$$A_\epsilon = I + \epsilon Z_A, \qquad B_\epsilon = I + \epsilon Z_B,$$

where $Z_A, Z_B$ are Gaussian random matrices with i.i.d. $\mathcal{N}(0, 1)$ entries, followed by row-wise normalization of $A_\epsilon$ to preserve identifiability. Each perturbed pair $(A_\epsilon, B_\epsilon)$ was used to transform the training and test datasets, and the reconstruction was then estimated using the procedure described in Section 4.

**Effective signal-to-noise ratio (SNR).** Rather than plotting the reconstruction error directly against $\epsilon$, we parameterize the x-axis in terms of an effective signal-to-noise ratio (SNR), defined as

$$\mathrm{SNR_F} := 20\log_{10}\left(\frac{\|X\|_F}{\|X - X_{\mathrm{noisy}}\|_F}\right),$$

where $X$ is the original image and $X_{\mathrm{noisy}}$ its transformed version. This reparametrization provides a scale-invariant measure of perturbation strength and directly reflects the degradation of signal energy in Frobenius norm.

**Results.** Figure 5 reports the reconstruction error as a function of the effective SNR for the Frobenius distance, while Figure 6 shows the analogous result for the element-wise maximum norm. In both cases, the distance between the original and corrected images (orange) is consistently below that between the original and noisy images (blue), confirming that the estimator effectively compensates for multiplicative perturbations in $A^*$ and $B^*$. The monotonic growth of both curves as SNR decreases quantifies the degradation rate, with the Frobenius error emphasizing global reconstruction quality and the max-norm capturing local, element-wise discrepancies.

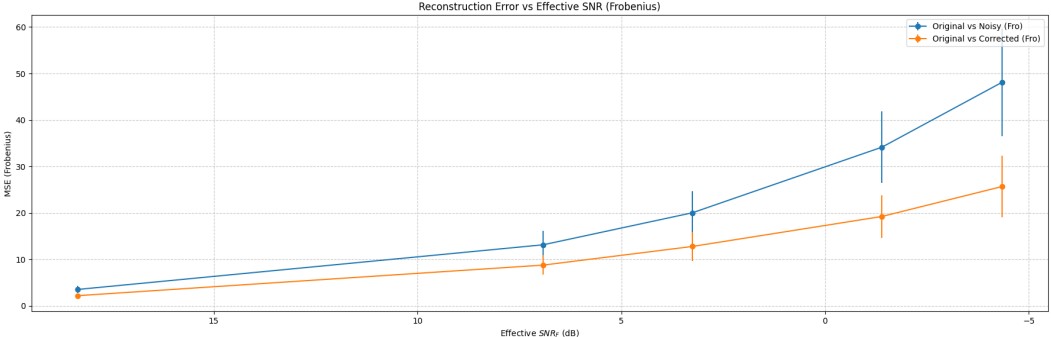

Figure 5: Reconstruction error versus effective SNR (Frobenius norm). Lower distances indicate improved correction quality. The x-axis is expressed in dB following the definition $\mathrm{SNR_F} = 20\log_{10}(\|X\|_F/\|X - X_{\mathrm{noisy}}\|_F)$.

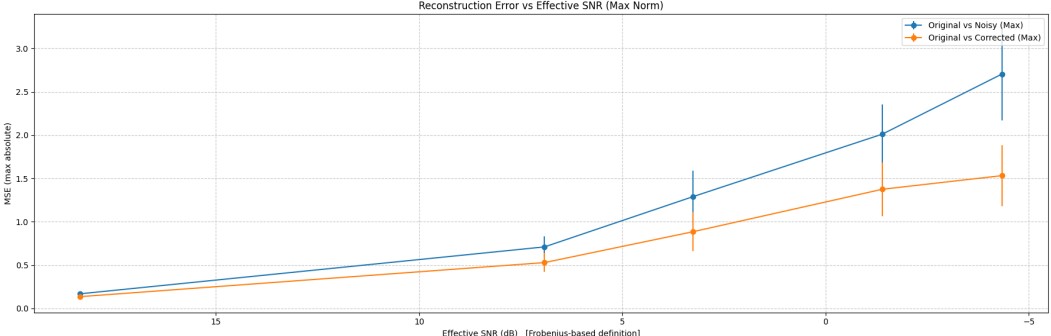

Figure 6: Reconstruction error versus effective SNR (element-wise maximum norm). This complementary metric highlights the preservation of fine-scale details under increasing noise.

Overall, the analysis confirms that the proposed correction procedure maintains stable performance across a broad range of noise intensities and that the reconstructed images preserve both global and local structure in accordance with the theoretical robustness guarantees.

## C Proofs

### C.1 Proofs of Section 2

We first analyze a simplified case of (2) where the design matrices $(X_t)_{t=1}^T$ are the elements of the canonical basis of $\mathbb{R}^{m \times q}$. In this setup we are given $T = mq$ observations $(X_t, M_t)_{t=1}^T$ where each predictor $X_t$ is a matrix with all entries set to zero except for a single entry equal to one. This setting is well studied in the vector case, typically referred to as a Sequence Model. Such models have been widely explored in the literature [44, 26]. By focusing on this simple scenario, we aim to uncover insights that can extend to more complex cases.

Formally, for $(k, l) \in [m] \times [q]$, let $M_{(k,l)}$ corresponds to the basis element $E_{(k,l)}$, where $E_{(k,l)}$ is the matrix with all entries equal to zero except for a one in the $k$-th row and $l$-th column. The model (2) can then be expressed as:

$$M_{(k,l)} = A^* E_{(k,l)} B^*, \tag{7}$$

which corresponds to (2) for $t = k + (l - 1)m$. The matrix $E_{(k,l)}$ acts as a selector, isolating the effect of the $k$-th row of $A^*$ and the $l$-th row of $B^*$.

**Lemma C.1.** *In the model* (7)*, the parameters $A^*$ and $B^*$ can be explicitly recovered from the observed matrices $M_{(k,l)}$ as follows:*

$$\forall (l, j) \in [q] \times [p] : [B^*]_{lj} = \sum_{k=1}^m [M_{(k,l)}]_{ij}, \ \forall i \in [n],$$

$$\forall (i, k) \in [n] \times [m] : [A^*]_{ik} = \frac{[M_{(k,l)}]_{ij}}{[B^*]_{lj}},$$

*for any $(l, j) \in [q] \times [p]$ such that $[B^*]_{lj} \neq 0$.*

*Proof of Lemma C.1.* From the model (7), we have

$$M_{(k,l)} = A^* E_{(k,l)} B^*,$$

where $E_{(k,l)}$ is the matrix with a single entry of 1 at the $(k, l)$-th position and zeros elsewhere. Expanding this relationship element-wise gives:

$$[M_{(k,l)}]_{ij} = [A^*]_{ik} \cdot [B^*]_{lj}, \ \forall (i, j, k, l) \in [n] \times [p] \times [m] \times [q].$$

Step 1: Recovery of $[B^*]_{lj}$.
Summing over $k \in [m]$ for fixed $l, j, i$, we observe that:

$$\sum_{k=1}^m [M_{(k,l)}]_{ij} = \sum_{k=1}^m [A^*]_{ik} \cdot [B^*]_{lj}.$$

Since the rows of $A^*$ have unit $L_1$-norm, it follows that:

$$[B^*]_{lj} = \sum_{k=1}^m [M_{(k,l)}]_{ij}, \ \forall (l, j, i) \in [q] \times [p] \times [n].$$

Step 2: Recovery of $[A^*]_{ik}$.
From the model equation, for each $(k, l)$, we isolate $[A^*]_{ik}$ by dividing $[M_{(k,l)}]_{ij}$ by $[B^*]_{lj}$, provided that $[B^*]_{lj} \neq 0$:

$$[A^*]_{ik} = \frac{[M_{(k,l)}]_{ij}}{[B^*]_{lj}}, \ \forall (i, k) \in [n] \times [m], \ \text{where } [B^*]_{lj} \neq 0.$$

$\square$

**Lemma C.2.** *In the model* (2)*, where the design matrices $X_t$ form a generating family of $\mathbb{R}^{m \times q}$, the unobserved matrix $\mathbb{C} \in \mathbb{R}^{mq \times np}$ satisfies the following equality:*

$$\mathbb{C} = \left(\mathbb{X}^\top \mathbb{X}\right)^{-1} \mathbb{X}^\top \mathbb{M}.$$

*Here, $\mathbb{M} := (vec(M_1), \ldots, vec(M_T))^\top \in \mathbb{R}^{T \times np}$ and $\mathbb{X} := (vec(X_1), \ldots, vec(X_T))^\top \in \mathbb{R}^{T \times mq}$*

*Proof of Lemma C.2.* We notice that $\mathbb{M} = \mathbb{X}\mathbb{C}$ by construction, which concludes the proof. □

*Proof of Proposition 2.2.* From Lemma C.2, the unobserved matrix $\mathbb{C}$ can be expressed as:

$$\mathbb{C} = \left(\mathbb{X}^{\top}\mathbb{X}\right)^{-1}\mathbb{X}^{\top}\mathbb{M}.$$

This means that $\mathbb{C}$ can be computed directly from the observations $(X_t, M_t)_{t \in [T]}$, provided that $\mathbb{X}^{\top}\mathbb{X}$ is invertible, which is ensured by the assumption that the design matrices $(X_t)_{t=1}^{T}$ form a generating family of $\mathbb{R}^{m \times q}$.

Step 1: Relating $\mathbb{C}$ to $A^*$ and $B^*$.
By the definition of $\mathbb{C}$, each entry $[\mathbb{C}]_{(k,l)}^{(i,j)}$ corresponds to:

$$[\mathbb{C}]_{(k,l)}^{(i,j)} = [A^*]_{ik} \cdot [B^*]_{lj},$$

for all $i \in [n]$, $k \in [m]$, $l \in [q]$, and $j \in [p]$. This bilinear structure allows us to recover $A^*$ and $B^*$ separately by exploiting their roles in this product.

Step 2: Recovering $B^*$.
To isolate $[B^*]_{lj}$, we sum $[\mathbb{C}]_{(k,l)}^{(i,j)}$ over all $k \in [m]$ for a fixed $l, j, i$. Specifically:

$$\sum_{k=1}^{m}[\mathbb{C}]_{(k,l)}^{(i,j)} = \sum_{k=1}^{m}\left([A^*]_{ik} \cdot [B^*]_{lj}\right).$$

Since the rows of $A^*$ satisfy the $L_1$-norm constraint (i.e., $\sum_{k=1}^{m}[A^*]_{ik} = 1$), the summation simplifies to:

$$\sum_{k=1}^{m}[\mathbb{C}]_{(k,l)}^{(i,j)} = [B^*]_{lj}.$$

This establishes the second equation in the proposition:

$$[B^*]_{lj} = \sum_{k=1}^{m}[\mathbb{C}]_{(k,l)}^{(i,j)}.$$

Step 3: Recovering $A^*$.
To isolate $[A^*]_{ik}$, we use the bilinear relationship:

$$[\mathbb{C}]_{(k,l)}^{(i,j)} = [A^*]_{ik} \cdot [B^*]_{lj}.$$

For a fixed $i, k, l, j$, we can solve for $[A^*]_{ik}$ provided that $[B^*]_{lj} \neq 0$:

$$[A^*]_{ik} = \frac{[\mathbb{C}]_{(k,l)}^{(i,j)}}{[B^*]_{lj}}.$$

This establishes the first equation in the proposition:

$$[A^*]_{ik} = \frac{[\mathbb{C}]_{(k,l)}^{(i,j)}}{[B^*]_{lj}}, \quad \text{for} \quad [B^*]_{lj} \neq 0.$$

□

*Proof of Corollary 2.4.* The result on $B^*$ follows immediately from Proposition 2.2. To prove the result on $A^*$, we need to prove that for any $n \in \mathbb{N}^*$, if there exist $(\alpha_1, \ldots, \alpha_n) \in \mathbb{R}^n$ and $(\beta_1, \ldots, \beta_n) \in \mathbb{R}^n$ such that:

$$\gamma_n := \frac{\alpha_1}{\beta_1} = \cdots = \frac{\alpha_n}{\beta_n},$$

then it follows that:

$$\gamma_n = \frac{\sum_{i=1}^{n}\alpha_i}{\sum_{j=1}^{n}\beta_j}.$$

Specifically, since the entries of $A^*$ can be expressed as different ratios, all being equal, applying this result to the equations satisfied by $A^*$ in Proposition 2.2 completes the proof.

We prove this result by induction on $n$.

Initially at $n = 1$ the result is trivially true.

Assume the statement holds at step $n$, i.e., if

$$\gamma_n := \frac{\alpha_1}{\beta_1} = \cdots = \frac{\alpha_n}{\beta_n},$$

then:

$$\gamma_n = \frac{\sum\limits_{i=1}^{n} \alpha_i}{\sum\limits_{j=1}^{n} \beta_j}.$$

We now prove that the statement holds at step $n + 1$.

Suppose:

$$\gamma_{n+1} := \frac{\alpha_1}{\beta_1} = \cdots = \frac{\alpha_{n+1}}{\beta_{n+1}}.$$

By the definition of $\gamma_{n+1}$ and the assumption of step $n$ being true, we can write:

$$\frac{\sum\limits_{i=1}^{n} \alpha_i}{\sum\limits_{j=1}^{n} \beta_j} = \gamma_{n+1} \quad \text{and} \quad \frac{\alpha_{n+1}}{\beta_{n+1}} = \gamma_{n+1}.$$

This allows to write:

$$\sum_{i=1}^{n} \alpha_i = \left( \sum_{j=1}^{n} \beta_j \right) \frac{\alpha_{n+1}}{\beta_{n+1}}.$$

Hence we deduce:

$$\frac{\sum\limits_{i=1}^{n} \alpha_i + \alpha_{n+1}}{\sum\limits_{j=1}^{n} \beta_j + \beta_{n+1}} = \frac{\left( \sum\limits_{j=1}^{n} \beta_j \right) \frac{\alpha_{n+1}}{\beta_{n+1}} + \alpha_{n+1}}{\sum\limits_{j=1}^{n} \beta_j + \beta_{n+1}},$$

$$= \alpha_{n+1} \cdot \frac{\frac{\left( \sum\limits_{j=1}^{n} \beta_j \right)}{\beta_{n+1}} + 1}{\sum\limits_{j=1}^{n} \beta_j + \beta_{n+1}}.$$

Finally multiplying the numerator by $\beta_{n+1}$ and dividing $\alpha_{n+1}$ by the same factor provides:

$$\frac{\sum\limits_{i=1}^{n} \alpha_i + \alpha_{n+1}}{\sum\limits_{j=1}^{n} \beta_j + \beta_{n+1}} = \frac{\alpha_{n+1}}{\beta_{n+1}} = \gamma_{n+1}.$$

Thus, the statement holds at step $n + 1$. $\qquad\square$

## C.2   Proofs of Section 3

*Proof of Lemma 3.1.* We first recall that, by construction, we have:

$$\widehat{\mathbb{C}} = \left( \mathbb{X}^\top \mathbb{X} \right)^{-1} \mathbb{X}^\top \mathbb{M} + \left( \mathbb{X}^\top \mathbb{X} \right)^{-1} \mathbb{X}^\top \mathbb{E} = \mathbb{C} + \mathbb{D}.$$

Next we prove the following, which will conclude:

$$\mathbb{D} \sim \mathcal{MN}_{mq \times np} \left( 0_{mq \times np}, \left( \mathbb{X}^\top \mathbb{X} \right)^{-1}, \Sigma_c \otimes \Sigma_r \right).$$

First we recall that the noise matrices $(E_t)_{t=1}^T$ are assumed to be independent and follow the same matrix normal distribution, namely $\mathcal{MN}_{n \times p} (0_{n \times p}, \Sigma_r, \Sigma_c)$. Hence by definition of the matrix normal distribution:

$$\mathrm{vec}(E_t) \overset{\mathrm{i.i.d.}}{\sim} \mathcal{N}_{np} \left( 0_{np}, \Sigma_c \otimes \Sigma_r \right).$$

This leads to derive the distribution of the matrix $\mathbb{E} \in \mathbb{R}^{T \times np}$ defined in (**??**) as follows:

$$\mathbb{E} \sim \mathcal{MN}_{T \times np} \left( 0_{T \times np}, I_{T \times T}, \Sigma_c \otimes \Sigma_r \right).$$

Finally, the matrix $\left( \mathbb{X}^\top \mathbb{X} \right)^{-1} \mathbb{X}^\top$ being of full rank $mq$ and $\left( \mathbb{X}^\top \mathbb{X} \right)^{-1}$ being symmetric, by definition of the matrix $\mathbb{D} \in \mathbb{R}^{mq \times np}$ and by property of affine transformations of matrix gaussian distributions, we conclude that

$$\mathbb{D} \sim \mathcal{MN}_{mq \times np} \left( 0_{T \times np}, \left( \mathbb{X}^\top \mathbb{X} \right)^{-1}, \Sigma_c \otimes \Sigma_r \right).$$

$\square$

*Proof of Theorem 3.3.* We start by deriving the finite-sampled inequalities on $\hat{B}$. From Lemma 3.1 and Assumption 3.2 we deduce that $\mathrm{vec}(\widehat{\mathbb{C}}) \sim \mathcal{N}_{mqnp} \left( \mathrm{vec}(\mathbb{C}), \dfrac{\Sigma_c \otimes \Sigma_r}{T} \right)$. As mentioned in the introduction $\Sigma_c \otimes \Sigma_r = \sigma^2 I_{np}$ which ensures that all entries of $\mathrm{vec}(\widehat{\mathbb{C}})$ are independent and follow the same Gaussian distribution. Hence for all $(k, l, i, j) \in [m] \times [q] \times [n] \times [p]$ we have $\widehat{\mathbb{C}}_{(k,l),(i,j)} - \mathbb{C}_{(k,l),(i,j)} \overset{i.i.d}{\sim} \mathcal{N} \left( 0, \dfrac{\sigma^2}{T} \right)$. Using the results from Proposition 2.2 and (3) lead to, for all $(l, j) \in [q] \times [p]$,

$$[\hat{B}]_{lj} - [B^*]_{lj} \overset{i.i.d}{\sim} \mathcal{N} \left( 0, \frac{m\sigma^2}{nT} \right). \tag{8}$$

Mill's inequality, Theorem E.1, ensures that for all $(l, j) \in [q] \times [p]$, for any $\epsilon > 0$,

$$\mathbb{P} \left[ \left| [\hat{B}]_{lj} - [B^*]_{lj} \right| > \epsilon \right] \leq \frac{\sigma}{\epsilon} \sqrt{\frac{2m}{nT\pi}} \exp \left( -\frac{nT\epsilon^2}{2m\sigma^2} \right).$$

The first inequality is then deduced by using a union bound.

The second inequality is immediately derived from the concentration of extreme singular values of Gaussian matrices with independent entries, see Corollary 5.35 in [46].

For the third inequality, we first note that $\|\hat{B} - B^*\|_F^2 = \sum_{l=1}^q \sum_{j=1}^p \left( [\hat{B}]_{lj} - [B^*]_{lj} \right)^2$ follows a $\dfrac{m\sigma^2}{nT} \cdot$ $\chi^2(pq)$ distribution. Then, Lemma 1 from [31] ensures that a random variable $Z$ following a $\chi^2(pq)$ distribution satisfies, for any $\epsilon > 0$:

$$\mathbb{P} \left( Z \geq pq + 2\sqrt{pq\epsilon} + 2\epsilon \right) \leq \exp(-\epsilon).$$

Hence we deduce that

$$\mathbb{P} \left( \|\hat{B} - B^*\|_F^2 \geq \frac{m\sigma^2}{nT} \left( pq + 2\sqrt{pq\epsilon} + 2\epsilon \right) \right) \leq \exp(-\epsilon).$$

We notice that, for any $\epsilon > 0$, the following inequality holds:

$$\frac{m\sigma^2}{nT} \left( pq + 2\sqrt{pq\epsilon} + 2\epsilon \right) \leq \frac{m\sigma^2}{nT} \left( 2pq + 3\epsilon \right)$$

It leads to:

$$\mathbb{P} \left( \|\hat{B} - B^*\|_F^2 \geq \frac{m\sigma^2}{nT} \left( 2pq + 3\epsilon \right) \right) \leq \exp(-\epsilon).$$

The stated result follows by a change of variable. $\square$

*Proof of Lemma 3.5.* Let us fix $(i, k) \in [n] \times [m]$. Using the definitions of $\hat{A}$ and $\tilde{A}$ from (4) and (3), respectively, we find that $[\tilde{A}]_{ik} = [\hat{A}]_{ik}$ if and only if $[\tilde{A}]_{ik} \in [0, 1]$. Hence $[\tilde{A}]_{ik} = [\hat{A}]_{ik}$ if and only if the event $\mathcal{A}$ holds, where:

$$\mathcal{A} := \left\{ 0 \leq \frac{\sum\limits_{j=1}^{p} \sum\limits_{l=1}^{q} \left[ \widehat{\mathbb{C}} \right]_{(k,l)}^{(i,j)}}{\sum\limits_{j=1}^{p} \sum\limits_{l=1}^{q} [\tilde{B}]_{lj}^{(i)}} \leq 1 \right\}.$$

This event is realized if and only if the event $\mathcal{A}_+$ or the event $\mathcal{A}_-$ holds, where:

$$\mathcal{A}_+ := \left\{ 0 \leq \sum\limits_{j=1}^{p} \sum\limits_{l=1}^{q} \left[ \widehat{\mathbb{C}} \right]_{(k,l)}^{(i,j)} \leq \sum\limits_{j=1}^{p} \sum\limits_{l=1}^{q} [\tilde{B}]_{lj}^{(i)} \right\},$$

and

$$\mathcal{A}_- := \left\{ \sum\limits_{j=1}^{p} \sum\limits_{l=1}^{q} [\tilde{B}]_{lj}^{(i)} \leq \sum\limits_{j=1}^{p} \sum\limits_{l=1}^{q} \left[ \widehat{\mathbb{C}} \right]_{(k,l)}^{(i,j)} \leq 0 \right\}.$$

Using Fréchet inequalities stated in Theorem E.2, we get:

$$\mathbb{P}[\mathcal{A}] \geq \max(\mathbb{P}[\mathcal{A}_+], \mathbb{P}[\mathcal{A}_-]).$$

Under Assumption 3.2, Lemma 3.1 gives:

$$\hat{\gamma}_{(k,i)} \sim \mathcal{N}\left( \gamma_{(k,i)}^*, \frac{\sigma^2}{pqT} \right), \tag{9}$$

where:

$$\hat{\gamma}_{(k,i)} := \frac{1}{pq} \sum\limits_{j=1}^{p} \sum\limits_{l=1}^{q} \left[ \widehat{\mathbb{C}} \right]_{(k,l)}^{(i,j)}$$

and

$$\gamma_{(k,i)}^* := \frac{1}{pq} \sum\limits_{j=1}^{p} \sum\limits_{l=1}^{q} [\mathbb{C}]_{(k,l),(i,j)}.$$

Under the same assumptions, as detailed in the proof of Theorem 3.3:

$$\hat{\beta}_i \sim \mathcal{N}\left( \beta^*, \frac{\sigma^2 m}{(n-1)pqT} \right), \tag{10}$$

where:

$$\hat{\beta}_i := \frac{1}{pq} \sum\limits_{j=1}^{p} \sum\limits_{l=1}^{q} \left[ \tilde{B} \right]_{lj}^{(i)} \quad \text{and} \quad \beta^* := \frac{1}{pq} \sum\limits_{j=1}^{p} \sum\limits_{l=1}^{q} [B^*]_{lj}.$$

By construction, $\hat{\gamma}_{(k,i)}$ and $\hat{\beta}_i$ are independent. In addition, as $\gamma_{(k,i)}^* = [A^*]_{ik}\beta^*$, Assumption 3.4 ensures that their expected values satisfy:

$$0 < \gamma_{(k,i)}^* < \beta^*.$$

Using the symmetry of the Gaussian distribution, we deduce:

$$\mathbb{P}[\mathcal{A}_+] \geq \mathbb{P}[\mathcal{A}_-] \quad \text{and} \quad \mathbb{P}[\mathcal{A}] \geq \mathbb{P}[\mathcal{A}_+].$$

The event $\mathcal{A}_+$ holds if $\mathcal{B}_1$ and $\mathcal{B}_2$ hold simultaneously, where:

$$\mathcal{B}_1 := \left\{ 0 \leq \hat{\gamma}_{(k,i)} \right\}, \quad \mathcal{B}_2 := \left\{ \hat{\gamma}_{(k,i)} \leq \hat{\beta}_i \right\}.$$

Using Fréchet inequalities, stated in Theorem E.2:

$$\mathbb{P}[\mathcal{A}_+] \geq \mathbb{P}[\mathcal{B}_1] + \mathbb{P}[\mathcal{B}_2] - 1.$$

We now bound from above the probability of the events $\mathcal{B}_1$ and $\mathcal{B}_2$ separately. For $\mathcal{B}_1$, we note:

$$\mathcal{B}_1 = \left\{ \gamma^*_{(k,i)} \geq \gamma^*_{(k,i)} - \hat{\gamma}_{(k,i)} \right\}.$$

The complementary event $\bar{\mathcal{B}}_1$ is:

$$\bar{\mathcal{B}}_1 = \left\{ \gamma^*_{(k,i)} \leq \gamma^*_{(k,i)} - \hat{\gamma}_{(k,i)} \right\}.$$

As we have $\gamma^*_{(k,i)} > 0$, using (9) and Mill's inequality stated in Theorem E.1 provides:

$$\mathbb{P}\left[\bar{\mathcal{B}}_1\right] \leq \frac{\sigma}{\gamma^*_{(k,i)}\sqrt{2pqT\pi}} \cdot \exp\left( -\frac{Tpq\left(\gamma^*_{(k,i)}\right)^2}{2\sigma^2} \right).$$

For $\mathcal{B}_2$ we note that:

$$\mathcal{B}_2 = \left\{ \beta^* - \gamma^*_{(k,i)} \geq (\beta^* - \gamma^*_{(k,i)}) - (\hat{\beta}_i - \hat{\gamma}_{(k,i)}) \right\}.$$

The complementary event $\bar{\mathcal{B}}_2$ is:

$$\bar{\mathcal{B}}_2 = \left\{ \beta^* - \gamma^*_{(k,i)} \leq (\beta^* - \gamma^*_{(k,i)}) - (\hat{\beta}_i - \hat{\gamma}_{(k,i)}) \right\}.$$

As we have $\beta^* - \gamma^*_{(k,i)} > 0$, using independence of $\hat{\gamma}_{(k,i)}$ and $\hat{\beta}_i$, results from (9), (10) and Mill's inequality ensures:

$$\mathbb{P}\left[\bar{\mathcal{B}}_2\right] \leq \frac{\sigma\sqrt{1+\frac{m}{n-1}}\exp\left( -\frac{Tpq\left(\beta^* - \gamma^*_{(k,i)}\right)^2}{2\sigma^2} \right)}{(\beta^* - \gamma^*_{(k,i)})\sqrt{2pqT\pi}}.$$

Finally, using:

$$\mathbb{P}\left[\mathcal{A}_+\right] \geq 1 - \mathbb{P}\left[\bar{\mathcal{B}}_1\right] - \mathbb{P}\left[\bar{\mathcal{B}}_2\right],$$

we deduce the result. $\qquad\square$

*Proof of Theorem 3.6.* Let us fix $(i,k) \in [n] \times [m]$. We work on the event $\mathcal{A} := \left\{ [\tilde{A}]_{ik} = [\hat{A}]_{ik} \right\}$. Using (3), we get:

$$\frac{1}{pq}[\tilde{A}]_{ik} \sum_{j=1}^{p}\sum_{l=1}^{q}[\tilde{B}]_{lj}^{(i)} = \frac{1}{pq}\sum_{j=1}^{p}\sum_{l=1}^{q}\left[\hat{\mathbb{C}}\right]_{(k,l)}^{(i,j)}.$$

Proposition 2.2 guarantees:

$$\frac{1}{pq}[A^*]_{ik} \sum_{j=1}^{p}\sum_{l=1}^{q}[B^*]_{lj} = \frac{1}{pq}\sum_{j=1}^{p}\sum_{l=1}^{q}[\mathbb{C}]_{(k,l),(i,j)}.$$

By definition of $\beta^*$ and using the reverse triangle's inequality we get:

$$|\beta^*| \cdot \left| [\tilde{A}]_{ik} - [A^*]_{ik} \right| \leq \left| \hat{\gamma}_{(k,i)} - \gamma^*_{(k,i)} \right| + \left| [\tilde{A}]_{ik} \right| \left| \hat{\beta}_i - \beta^* \right|,$$

where:

$$\hat{\beta}_i := \frac{1}{pq}\sum_{j=1}^{p}\sum_{l=1}^{q}\left[\tilde{B}\right]_{lj}^{(i)}, \ \hat{\gamma}_{(k,i)} := \frac{1}{pq}\sum_{j=1}^{p}\sum_{l=1}^{q}\left[\hat{\mathbb{C}}\right]_{(k,l)}^{(i,j)}$$

and

$$\gamma^*_{(k,i)} := \frac{1}{pq}\sum_{j=1}^{p}\sum_{l=1}^{q}[\mathbb{C}]_{(k,l),(i,j)}.$$

Under Assumption 3.2, (9) and (10) hold. Using (9) and Mill's inequality stated in Theorem E.1, we get for any $\epsilon > 0$:

$$\mathbb{P}\left[\left|\hat{\gamma}_{(k,i)} - \gamma^*_{(k,i)}\right| > \epsilon\right] \leq \frac{\sigma\sqrt{2}}{\epsilon\sqrt{pqT\pi}}\exp\left(-\frac{\epsilon^2 pqT}{2\sigma^2}\right).$$

Using (10) and Mill's inequality, we find for any $\epsilon > 0$:

$$\mathbb{P}\left[\left|\hat{\beta}_i - \beta^*\right| > \epsilon\right] \leq \frac{\sigma\sqrt{2m}}{\epsilon\sqrt{pqT\dot{n}\pi}}\exp\left(-\frac{\epsilon^2 pqT\dot{n}}{2m\sigma^2}\right),$$

where $\dot{n} := (n-1)$.

Finally, on the event $\mathcal{A}$, we have $\left|[\tilde{A}]_{ik}\right| \leq 1$. We then use the result from Lemma 3.5, Fréchet inequality stated in Theorem E.2, and we conclude with a union bound. $\qquad\square$

*Proof of Theorem 3.7.* For all $(l, j) \in [q] \times [p]$, (8) ensures that:

$$[\hat{B}]_{lj} = [B^*]_{lj} + \epsilon_{lj},$$

where:

$$\epsilon_{lj} \overset{i.i.d.}{\sim} \mathcal{N}\left(0, \frac{m\sigma^2}{nT}\right).$$

From the results in Appendix D, we obtain that the event $\mathcal{A}$ holds with probability at least $1 - \delta$ where $\mathcal{A}$ is the event

$$\max_{\substack{l\in[q]\\j\in[p]}} |\epsilon_{lj}| \leq \sigma\sqrt{\frac{2m}{nT}}\left(\sqrt{\log(2pq)} + \sqrt{\log\left(\frac{2}{\delta}\right)}\right).$$

We recall the definition of the threshold:

$$\tau := \sigma\sqrt{\frac{2m}{nT}}\left(\sqrt{\log(2pq)} + \sqrt{\log\left(\frac{2}{\delta}\right)}\right).$$

On the event $\mathcal{A}$, we observe the following:

- If $|[\hat{B}]_{lj}| > 2\tau$, then:
$$|[B^*]_{lj}| \geq |[\hat{B}]_{lj}| - |\epsilon_{lj}| > \tau.$$

- If $|[\hat{B}]_{lj}| \leq 2\tau$, then:
$$|[B^*]_{lj}| \leq |[\hat{B}]_{lj}| + |\epsilon_{lj}| \leq 3\tau.$$

From these observations, we deduce:

$$\left|[\hat{B}^S]_{lj} - [B^*]_{lj}\right| = \begin{cases} \left|[\hat{B}]_{lj} - [B^*]_{lj}\right|, & \text{if } |[\hat{B}]_{lj}| > 2\tau, \\ |[B^*]_{lj}|, & \text{if } |[\hat{B}]_{lj}| \leq 2\tau. \end{cases}$$

Rewriting this equality with indicator functions, we have:

$$\left|[\hat{B}^S]_{lj} - [B^*]_{lj}\right| = |\epsilon_{lj}|\mathbb{1}\left(|[\hat{B}]_{lj}| > 2\tau\right),$$
$$+ |[B^*]_{lj}|\mathbb{1}\left(|[\hat{B}]_{lj}| \leq 2\tau\right).$$

This leads to the bound:

$$\left|[\hat{B}^S]_{lj} - [B^*]_{lj}\right| \leq \tau\mathbb{1}\left(|[B^*]_{lj}| > \tau\right),$$
$$+ 3\tau\mathbb{1}\left(|[B^*]_{lj}| \leq 3\tau\right).$$

Finally, we conclude:

$$\left| [\hat{B}^S]_{lj} - [B^*]_{lj} \right| \leq 4 \min \left( \tau, |[B^*]_{lj}| \right).$$

By definition of the Frobenius norm, we obtain:

$$\left\| \hat{B}^S - B^* \right\|_F^2 \leq 16 \left\| B^* \right\|_0 \tau^2.$$

This proves the first point.

For the second point, on the event $\mathcal{A}$ and under the stated assumption, we observe the following:

- If $|[B^*]_{lj}| \neq 0$, then $|[B^*]_{lj}| > 3\tau$. This provides the bound:

$$|[\hat{B}]_{lj}| = |[B^*]_{lj} + \epsilon_{lj}| > 2\tau.$$

  Hence $|[\hat{B}^S]_{lj}| \neq 0$.

- If $|[\hat{B}^S]_{lj}| \neq 0$, then $|[\hat{B}]_{lj}| > 2\tau$. This provides the bound:

$$|[B^*]_{lj}| \geq |[\hat{B}]_{lj}| - |\epsilon_{lj}| \geq \tau.$$

  Hence $|[B^*]_{lj}| \neq 0$.

$\square$

*Proof of Theorem 3.8.* For all $(i, k) \in [n] \times [m]$, (9) ensures that:

$$[\hat{\gamma}]_{ik} = [\gamma^*]_{ik} + \epsilon_{ik},$$

where:

$$\gamma^*_{ik} := \frac{1}{pq} \sum_{j=1}^{p} \sum_{l=1}^{q} [\mathbb{C}]_{(k,l),(i,j)}$$

and

$$\epsilon_{ik} \overset{\text{i.i.d.}}{\sim} \mathcal{N}\left( 0, \frac{\sigma^2}{pqT} \right).$$

Proposition 2.2 ensures that

$$\gamma^*_{ik} = [A^*]_{ik} \beta^*.$$

From the results in Appendix D, we obtain that the event $\mathcal{A}$ holds with probability at least $1 - \delta$ where $\mathcal{A}$ is the event

$$\max_{\substack{i \in [n] \\ k \in [m]}} |\epsilon_{ik}| \leq \sigma \sqrt{\frac{2}{Tpq}} \left( \sqrt{\log(2nm)} + \sqrt{\log\left( \frac{2}{\delta} \right)} \right).$$

We recall the definition of the threshold:

$$\tau := \sigma \sqrt{\frac{2}{Tpq}} \left( \sqrt{\log(2nm)} + \sqrt{\log\left( \frac{2}{\delta} \right)} \right).$$

On the event $\mathcal{A}$, we observe the following:

- If $|\hat{\gamma}_{ik}| > 2\tau$, then:
$$|\gamma^*_{ik}| \geq |\hat{\gamma}_{ik}| - |\epsilon_{ik}| > \tau.$$
  Thus we have $[A^*]_{ik} \neq 0$ and because $[A^*]_{ik}$ is bounded from above by 1 we also have $\beta^* \geq \tau$.

- If $|\hat{\gamma}_{ik}| \leq 2\tau$, then:
$$|\gamma^*_{ik}| \leq |\hat{\gamma}_{ik}| + |\epsilon_{ik}| \leq 3\tau.$$

From these observations, we deduce:

$$\left|[\hat{A}^S]_{ik} - [A^*]_{ik}\right| = \begin{cases} \left|[\hat{A}]_{ik} - [A^*]_{ik}\right|, & \text{if } |\hat{\gamma}_{ik}| > 2\tau, \\ |[A^*]_{ik}|, & \text{if } |\hat{\gamma}_{ik}| \le 2\tau. \end{cases}$$

Rewriting this with indicator functions and using the previously stated implications, we have:

$$\left|[\hat{A}^S]_{ik} - [A^*]_{ik}\right| \le \left|[\hat{A}]_{ik} - [A^*]_{ik}\right| \mathbb{1}\left(|\gamma_{ik}^*| > \tau\right),$$
$$+ |[A^*]_{ik}| \mathbb{1}\left(|\gamma_{ik}^*| \le 3\tau\right).$$

Multiplying all terms by $|\beta^*|$ leads to the bound:

$$|\beta^*| \cdot |[\hat{A}^S]_{lj} - [A^*]_{lj}| \le |\beta^*| \left|[\hat{A}]_{ik} - [A^*]_{ik}\right| \mathbb{1}\left(|\gamma_{ik}^*| > \tau\right),$$
$$+ |\gamma_{ik}^*| \mathbb{1}\left(|\gamma_{ik}^*| \le 3\tau\right).$$

Using the equality

$$|\beta^*| \left|[\hat{A}]_{ik} - [A^*]_{ik}\right| = \left|(\beta^* - \hat{\beta})[\hat{A}]_{ik} + \hat{\beta}[\hat{A}]_{ik} - \beta^*[A^*]_{ik}\right|$$

and the triangle inequality leads to

$$|\beta^*| \cdot |[\hat{A}^S]_{ik} - [A^*]_{ik}| \le (M_{ik}^{(1)} + M_{ik}^{(2)}) \mathbb{1}\left(|\gamma_{ik}^*| > \tau\right),$$
$$+ |\gamma_{ik}^*| \mathbb{1}\left(|\gamma_{ik}^*| \le 3\tau\right),$$

where

$$M_{ik}^{(1)} = |\beta^* - \hat{\beta}| \left|[\hat{A}]_{ik}\right|$$

and

$$M_{ik}^{(2)} = \left|\hat{\beta}[\hat{A}]_{ik} - \beta^*[A^*]_{ik}\right|.$$

Moreover we notice that

$$M_{ik}^{(2)} = |\hat{\gamma}_{ik} - \gamma_{ik}^*|.$$

Equations (9) and (10), together with Mill's inequality, stated in Theorem E.1, provide for any $t > 0$:

$$\mathbb{P}\left[\left|\beta^* - \hat{\beta}\right| > t\right] \le \frac{\sigma\sqrt{2m}}{t\sqrt{npqT\pi}} \exp\left(-\frac{Tpqt^2}{2\sigma^2 m}\right)$$

and

$$\mathbb{P}\left[|\gamma_{ik}^* - \hat{\gamma}_{ik}| > t\right] \le \frac{\sigma\sqrt{2}}{t\sqrt{pqT\pi}} \exp\left(-\frac{Tpqt^2}{2\sigma^2}\right).$$

Finally, we get for any $\delta \in (0,1)$ with probability at least $1 - \delta$:

$$|\beta^*| \cdot |[\hat{A}^S]_{ik} - [A^*]_{ik}| \le 2t_\delta \mathbb{1}\left(A_{ik}^* \ne 0\right) \mathbb{1}\left(|\beta^*| \ge \tau\right)$$
$$+ |\gamma_{ik}^*| \mathbb{1}\left(|\gamma_{ik}^*| \le 3\tau\right).$$

Dividing by $|\beta^*|^{-1}$ and using that $\gamma_{ik}^* = [A^*]_{ik}\beta^*$ ensures that with probability at least $1 - \delta$:

$$|[\hat{A}^S]_{ik} - [A^*]_{ik}| \le 2t_\delta|\beta^*|^{-1} \mathbb{1}\left(A_{ik}^* \ne 0\right) \mathbb{1}\left(|\beta^*|^{-1} \le \tau^{-1}\right)$$
$$+ [A^*]_{ik} \mathbb{1}\left([A^*]_{ik} \le 3\tau|\beta^*|^{-1}\right).$$

Finally we get with probability at least $1 - \delta$:

$$|[\hat{A}^S]_{ik} - [A^*]_{ik}| \le |\beta^*|^{-1} \mathbb{1}\left(A_{ik}^* \ne 0\right)(2t_\delta + 3\tau).$$

By definition of the Frobenius norm, we obtain with probability at least $1 - \delta$:

$$\left\|\hat{A}^S - A^*\right\|_F^2 \le |\beta^*|^{-2} \|A^*\|_0 (2t_\delta + 3\tau)^2.$$

For the second point, on the event $\mathcal{A}$ and under the stated assumptions, we observe the following:

- If $|[A^*]_{ik}| \neq 0$, then $|[A^*]_{ik}| > 3\tau$. Using the equality $\gamma_{ik}^* = \beta^* [A^*]_{ik}$ and the assumption on the bound satisfied by $\beta^*$ provide:
$$|\hat{\gamma}_{ik}| = |\gamma_{ik}^* + \epsilon_{ik}| > 3\beta^*\tau - \tau > 0.$$

  Hence $|[\hat{A}^S]_{ik}| \neq 0$.

- If $|[\hat{A}^S]_{ik}| \neq 0$, then $|\hat{\gamma}_{ik}| > 2\tau$. This provides the bound:
$$|\gamma_{lj}^*| \geq |\hat{\gamma}_{lj}| - |\epsilon_{lj}| \geq \tau.$$

  Hence $|\gamma_{lj}^*| \neq 0$ and thus $[A^*]_{ik} \neq 0$.

$\square$

## D   Tails of the Maximum of Absolute Gaussian Variables

In this appendix, adapted from [45], we analyze the upper and lower tails of the random variable $\max_{1 \leq i \leq n} |X_i|$, where $X_1, \ldots, X_n$ are independent and identically distributed (i.i.d.) random variables following a standard normal distribution $\mathcal{N}(0, 1)$. Specifically, we aim to establish bounds for the upper and lower tails of this random variable with high probability, which is crucial for understanding the behavior of maxima in Gaussian settings.

We begin by stating the main result.

**Theorem D.1.** *Fix $\delta \in (0, 1)$, and let $X_1, \ldots, X_n$ be i.i.d. $\mathcal{N}(0, 1)$. With probability at least $1 - \delta$,*

$$\sqrt{\frac{\pi}{2}} \sqrt{\log(n/2) - \log\log(2/\delta)} \leq \max_{1 \leq i \leq n} |X_i|,$$
$$\max_{1 \leq i \leq n} |X_i| \leq \sqrt{2} \left( \sqrt{\log(2n)} + \sqrt{\log(2/\delta)} \right).$$

The asymmetry in the tails arises from the fact that $\max_{1 \leq i \leq n} |X_i|$ is bounded below by zero almost surely, it is not bounded above by any fixed constant. To establish the theorem, we separately analyze the upper and lower tails and combine these results with a union bound.

### D.1   Upper Tail

The upper tail is analyzed using concentration results for the suprema of Gaussian processes. The key tool is Talagrand's concentration inequality, Lemma 2.10.6 in [42], stated in a simpler version as follows:

**Lemma D.2.** *Consider $T \subseteq \mathbb{R}^n$ and let $g \sim \mathcal{N}(0, I)$. Then, for any $u > 0$,*

$$\mathbb{P}\left\{ \sup_{t \in T} \langle t, g \rangle - \mathbb{E}\left[ \sup_{t \in T} \langle t, g \rangle \right] > u \right\} \leq \exp\left\{ -\frac{u^2}{2s^2} \right\},$$

*where $s := \sup_{t \in T} \mathbb{E}[\langle t, g \rangle^2]^{1/2}$.*

To apply this inequality, we consider $T = \{e_1, \ldots, e_n, -e_1, \ldots, -e_n\}$, where $e_i$ is the $i$-th canonical basis vector in $\mathbb{R}^n$. For $g \sim \mathcal{N}(0, I)$, this gives

$$\sup_{t \in T} \langle t, g \rangle = \max_{1 \leq i \leq n} |g_i|, \quad \text{and} \quad \mathbb{E}[\langle t, g \rangle^2] = \mathbb{E}[g_i^2] = 1.$$

Applying Lemma D.2, we obtain, for any $\delta > 0$:

$$\mathbb{P}\left( \max_{1 \leq i \leq n} |X_i| > \mathbb{E}\left[ \max_{1 \leq i \leq n} |X_i| \right] + \sqrt{2\log(1/\delta)} \right) \leq \delta.$$

Thus, with probability at least $1 - \delta$:

$$\max_{1 \leq i \leq n} |X_i| \leq \mathbb{E}\left[ \max_{1 \leq i \leq n} |X_i| \right] + \sqrt{2\log(1/\delta)}. \tag{11}$$

The next step is to bound $\mathbb{E}\left[ \max_{1 \leq i \leq n} |X_i| \right]$ from above.

## D.2 Expected Maximum of Gaussian Variables

We now bound $\mathbb{E}\left[\max_{1 \leq i \leq n} |X_i|\right]$.

**Proposition D.3.** *Let $Z_1, \ldots, Z_n$ be $n$ random variables (not necessarily independent) with marginal distribution $\mathcal{N}(0, 1)$. Then,*

$$\mathbb{E}\left[\max_{1 \leq i \leq n} Z_i\right] \leq \sqrt{2 \log n}.$$

*Proof.* Fix any $\lambda > 0$. Observe that

$$\mathbb{E}\left[e^{\lambda \max_{1 \leq i \leq n} Z_i}\right] \leq \sum_{i=1}^{n} \mathbb{E}\left[e^{\lambda Z_i}\right] = n e^{\lambda^2/2}.$$

Taking the logarithm of both sides,

$$\log \mathbb{E}\left[e^{\lambda \max_{1 \leq i \leq n} Z_i}\right] \leq \log n + \frac{\lambda^2}{2}.$$

Applying Jensen's inequality,

$$\lambda \mathbb{E}\left[\max_{1 \leq i \leq n} Z_i\right] \leq \log \mathbb{E}\left[e^{\lambda \max_{1 \leq i \leq n} Z_i}\right] \leq \log n + \frac{\lambda^2}{2}.$$

Dividing through by $\lambda$,

$$\mathbb{E}\left[\max_{1 \leq i \leq n} Z_i\right] \leq \frac{\log n}{\lambda} + \frac{\lambda}{2}.$$

Optimizing by setting $\lambda = \sqrt{2 \log n}$, we conclude

$$\mathbb{E}\left[\max_{1 \leq i \leq n} Z_i\right] \leq \sqrt{2 \log n}.$$

$\square$

For $X_1, \ldots, X_n$ i.i.d. following a standard normal distribution $\mathcal{N}(0, 1)$, considering the $2n$ variables $(X_1, \ldots, X_n, -X_1, \ldots, -X_n)$, we have:

$$\mathbb{E}\left[\max_{1 \leq i \leq n} |X_i|\right] \leq \sqrt{2 \log(2n)}.$$

Substituting this into (11) completes the proof of the upper tail.

## D.3 Lower Bound

We now analyze the lower tail of $\max_{1 \leq i \leq n} |X_i|$. Fix a positive $\tau > 0$. Then,

$$\Pr\left\{\max_{1 \leq i \leq n} |X_i| \leq \tau\right\} = \Pr\left(|X_1| \leq \tau, \ldots, |X_n| \leq \tau\right).$$

Using the independence of $X_1, \ldots, X_n$, we get:

$$\Pr\left\{\max_{1 \leq i \leq n} |X_i| \leq \tau\right\} = \prod_{i=1}^{n} \Pr\left(|X_i| \leq \tau\right)$$

The Gauss error function, defined as

$$\operatorname{erf} : z \mapsto \frac{2}{\sqrt{\pi}} \int_0^z e^{-t^2} \, dt,$$

allows then to write:

$$\Pr\left\{\max_{1 \leq i \leq n} |X_i| \leq \tau\right\} = \operatorname{erf}\left(\frac{\tau}{\sqrt{2}}\right)^n$$

The inequality $\mathrm{erf}(x)^2 \leq 1 - e^{-4x^2/\pi}$, which holds for all $x \geq 0$, provides:

$$\Pr\left\{\max_{1\leq i \leq n}|X_i| \leq \tau\right\} \leq \left(1 - e^{-\frac{2}{\pi}\tau^2}\right)^{n/2}$$

Finally, the inequality $1 - x \leq e^{-x}$, which holds for all $x \in \mathbb{R}$ ensures:

$$\Pr\left\{\max_{1\leq i \leq n}|X_i| \leq \tau\right\} \leq \exp\left(-\frac{n}{2}e^{-\frac{2}{\pi}\tau^2}\right).$$

To derive the lower bound, set

$$\exp\left(-\frac{n}{2}e^{-\frac{2}{\pi}\tau^2}\right) = \delta,$$

and solve for $\tau$. We find that with probability at least $1 - \delta$,

$$\max_{1\leq i \leq n}|X_i| \geq \sqrt{\frac{\pi}{2}\log(n/2) - \frac{\pi}{2}\log\log(1/\delta)}.$$

# E    Probability bounds and inequalities

## E.1    Mill's inequality

**Theorem E.1** (Mill's Inequality). *Let $X$ be a Gaussian random variable with mean $\mu$ and variance $\sigma^2$. Then, for any $t > 0$, the following inequality holds:*

$$\mathbb{P}(X - \mu > t) \leq \frac{\sigma}{t\sqrt{2\pi}}e^{-\frac{t^2}{2\sigma^2}}.$$

*By symmetry, we also have:*

$$\mathbb{P}(X - \mu < -t) \leq \frac{\sigma}{t\sqrt{2\pi}}e^{-\frac{t^2}{2\sigma^2}}.$$

*Proof.* A proof of this theorem can be found in [37]. $\qquad\qquad\square$

## E.2    Fréchet inequalities

**Theorem E.2** (Fréchet inequalities). *Let $A$ and $B$ be two events. The probability of their intersection satisfies:*

$$\max(0, \mathbb{P}(A) + \mathbb{P}(B) - 1) \leq \mathbb{P}(A \cap B) \leq \min(\mathbb{P}(A), \mathbb{P}(B)).$$

*The probability of their union satisfies:*

$$\max(\mathbb{P}(A), \mathbb{P}(B)) \leq \mathbb{P}(A \cup B) \leq \min(1, \mathbb{P}(A) + \mathbb{P}(B)).$$

# NeurIPS Paper Checklist

1. **Claims**

   Question: Do the main claims made in the abstract and introduction accurately reflect the paper's contributions and scope?

   Answer: [Yes]

   Justification: The main claims in the abstract and introduction are carefully aligned with the paper's actual contributions, as detailed in Section 1.3. These claims are neither overstated nor misleading. They accurately describe the scope of the work, including the introduction of optimization-free estimators, non-asymptotic error bounds, and the applicability to matrix-valued regression problems. The theoretical and empirical results presented in the core sections fully support these contributions.

   Guidelines:

   - The answer NA means that the abstract and introduction do not include the claims made in the paper.
   - The abstract and/or introduction should clearly state the claims made, including the contributions made in the paper and important assumptions and limitations. A No or NA answer to this question will not be perceived well by the reviewers.
   - The claims made should match theoretical and experimental results, and reflect how much the results can be expected to generalize to other settings.
   - It is fine to include aspirational goals as motivation as long as it is clear that these goals are not attained by the paper.

2. **Limitations**

   Question: Does the paper discuss the limitations of the work performed by the authors?

   Answer: [Yes]

   Justification: The limitations of the work are discussed in Appendix A, where we explicitly address the simplifying assumptions made in the theoretical analysis, such as the use of isotropic Gaussian noise, and explain how the results could be extended under more general settings. We also acknowledge the technical challenges involved in such extensions and provide references to the appropriate concentration inequalities required for handling non-isotropic noise or more complex covariance structures.

   Guidelines:

   - The answer NA means that the paper has no limitation while the answer No means that the paper has limitations, but those are not discussed in the paper.
   - The authors are encouraged to create a separate "Limitations" section in their paper.
   - The paper should point out any strong assumptions and how robust the results are to violations of these assumptions (e.g., independence assumptions, noiseless settings, model well-specification, asymptotic approximations only holding locally). The authors should reflect on how these assumptions might be violated in practice and what the implications would be.
   - The authors should reflect on the scope of the claims made, e.g., if the approach was only tested on a few datasets or with a few runs. In general, empirical results often depend on implicit assumptions, which should be articulated.
   - The authors should reflect on the factors that influence the performance of the approach. For example, a facial recognition algorithm may perform poorly when image resolution is low or images are taken in low lighting. Or a speech-to-text system might not be used reliably to provide closed captions for online lectures because it fails to handle technical jargon.
   - The authors should discuss the computational efficiency of the proposed algorithms and how they scale with dataset size.
   - If applicable, the authors should discuss possible limitations of their approach to address problems of privacy and fairness.

- While the authors might fear that complete honesty about limitations might be used by reviewers as grounds for rejection, a worse outcome might be that reviewers discover limitations that aren't acknowledged in the paper. The authors should use their best judgment and recognize that individual actions in favor of transparency play an important role in developing norms that preserve the integrity of the community. Reviewers will be specifically instructed to not penalize honesty concerning limitations.

3. **Theory assumptions and proofs**

   Question: For each theoretical result, does the paper provide the full set of assumptions and a complete (and correct) proof?

   Answer: [Yes]

   Justification: Each theoretical result in the paper is stated with a complete and explicit set of assumptions. The main assumptions are clearly outlined, and each theorem or lemma is accompanied by a full proof provided in Appendix C. The proofs are rigorous, and where simplifications are made (e.g., assuming isotropic noise), we clarify this explicitly and discuss how the results can be extended under milder assumptions in Appendix A.

   Guidelines:

   - The answer NA means that the paper does not include theoretical results.
   - All the theorems, formulas, and proofs in the paper should be numbered and cross-referenced.
   - All assumptions should be clearly stated or referenced in the statement of any theorems.
   - The proofs can either appear in the main paper or the supplemental material, but if they appear in the supplemental material, the authors are encouraged to provide a short proof sketch to provide intuition.
   - Inversely, any informal proof provided in the core of the paper should be complemented by formal proofs provided in appendix or supplemental material.
   - Theorems and Lemmas that the proof relies upon should be properly referenced.

4. **Experimental result reproducibility**

   Question: Does the paper fully disclose all the information needed to reproduce the main experimental results of the paper to the extent that it affects the main claims and/or conclusions of the paper (regardless of whether the code and data are provided or not)?

   Answer: [Yes]

   Justification: The paper provides a detailed description of the experimental setup, including the data generation process, model parameters, and evaluation metrics used in all simulations. Each figure in the results section is directly tied to a clearly defined experimental protocol. The code is included in the supplementary material and all information necessary to reproduce the results relevant to the main claims and conclusions is fully disclosed. The experiments serve to validate the theoretical findings, and no critical step is omitted in their description.

   Guidelines:

   - The answer NA means that the paper does not include experiments.
   - If the paper includes experiments, a No answer to this question will not be perceived well by the reviewers: Making the paper reproducible is important, regardless of whether the code and data are provided or not.
   - If the contribution is a dataset and/or model, the authors should describe the steps taken to make their results reproducible or verifiable.
   - Depending on the contribution, reproducibility can be accomplished in various ways. For example, if the contribution is a novel architecture, describing the architecture fully might suffice, or if the contribution is a specific model and empirical evaluation, it may be necessary to either make it possible for others to replicate the model with the same dataset, or provide access to the model. In general. releasing code and data is often one good way to accomplish this, but reproducibility can also be provided via detailed instructions for how to replicate the results, access to a hosted model (e.g., in the case of a large language model), releasing of a model checkpoint, or other means that are appropriate to the research performed.

- While NeurIPS does not require releasing code, the conference does require all submissions to provide some reasonable avenue for reproducibility, which may depend on the nature of the contribution. For example
    (a) If the contribution is primarily a new algorithm, the paper should make it clear how to reproduce that algorithm.
    (b) If the contribution is primarily a new model architecture, the paper should describe the architecture clearly and fully.
    (c) If the contribution is a new model (e.g., a large language model), then there should either be a way to access this model for reproducing the results or a way to reproduce the model (e.g., with an open-source dataset or instructions for how to construct the dataset).
    (d) We recognize that reproducibility may be tricky in some cases, in which case authors are welcome to describe the particular way they provide for reproducibility. In the case of closed-source models, it may be that access to the model is limited in some way (e.g., to registered users), but it should be possible for other researchers to have some path to reproducing or verifying the results.

5. **Open access to data and code**

   Question: Does the paper provide open access to the data and code, with sufficient instructions to faithfully reproduce the main experimental results, as described in supplemental material?

   Answer: [Yes]

   Justification: The full implementation is provided as a Jupyter notebook in the supplementary material, including all code and data needed to reproduce the main experimental results. The notebook contains clear documentation and step-by-step instructions, ensuring that the simulations and figures presented in the paper can be faithfully reproduced. This supports the transparency and reproducibility of the empirical claims made in the work. See `https://github.com/nayelbettache/BMLR`.

   Guidelines:

   - The answer NA means that paper does not include experiments requiring code.
   - Please see the NeurIPS code and data submission guidelines (`https://nips.cc/public/guides/CodeSubmissionPolicy`) for more details.
   - While we encourage the release of code and data, we understand that this might not be possible, so "No" is an acceptable answer. Papers cannot be rejected simply for not including code, unless this is central to the contribution (e.g., for a new open-source benchmark).
   - The instructions should contain the exact command and environment needed to run to reproduce the results. See the NeurIPS code and data submission guidelines (`https://nips.cc/public/guides/CodeSubmissionPolicy`) for more details.
   - The authors should provide instructions on data access and preparation, including how to access the raw data, preprocessed data, intermediate data, and generated data, etc.
   - The authors should provide scripts to reproduce all experimental results for the new proposed method and baselines. If only a subset of experiments are reproducible, they should state which ones are omitted from the script and why.
   - At submission time, to preserve anonymity, the authors should release anonymized versions (if applicable).
   - Providing as much information as possible in supplemental material (appended to the paper) is recommended, but including URLs to data and code is permitted.

6. **Experimental setting/details**

   Question: Does the paper specify all the training and test details (e.g., data splits, hyperparameters, how they were chosen, type of optimizer, etc.) necessary to understand the results?

   Answer: [Yes]

   Justification: For the synthetic experiments, all parameters, such as sample size, matrix dimensions, noise levels, and distributional assumptions, are fully specified. The estimators

are closed-form and do not involve optimization or hyperparameter tuning, so no additional training procedure is required. For the real-data experiments on CIFAR-10, the paper uses the standard train/test split provided by the dataset's official loader, which is explicitly mentioned in the text. No additional tuning or fine-tuning is performed, and all relevant details are disclosed to ensure full understanding of the results.

Guidelines:

- The answer NA means that the paper does not include experiments.
- The experimental setting should be presented in the core of the paper to a level of detail that is necessary to appreciate the results and make sense of them.
- The full details can be provided either with the code, in appendix, or as supplemental material.

7. **Experiment statistical significance**

Question: Does the paper report error bars suitably and correctly defined or other appropriate information about the statistical significance of the experiments?

Answer: [Yes]

Justification: For the simulated data, the paper evaluates performance across a wide range of parameter settings (e.g., sample size, matrix dimensions, noise levels), which helps mitigate the effects of randomness and ensures that the empirical results robustly reflect the theoretical predictions. This systematic variation serves as a form of sensitivity analysis. For the real-world CIFAR-10 experiments, the paper explicitly reports error bars to convey the variability of the estimators and support the statistical significance of the results.

Guidelines:

- The answer NA means that the paper does not include experiments.
- The authors should answer "Yes" if the results are accompanied by error bars, confidence intervals, or statistical significance tests, at least for the experiments that support the main claims of the paper.
- The factors of variability that the error bars are capturing should be clearly stated (for example, train/test split, initialization, random drawing of some parameter, or overall run with given experimental conditions).
- The method for calculating the error bars should be explained (closed form formula, call to a library function, bootstrap, etc.)
- The assumptions made should be given (e.g., Normally distributed errors).
- It should be clear whether the error bar is the standard deviation or the standard error of the mean.
- It is OK to report 1-sigma error bars, but one should state it. The authors should preferably report a 2-sigma error bar than state that they have a 96% CI, if the hypothesis of Normality of errors is not verified.
- For asymmetric distributions, the authors should be careful not to show in tables or figures symmetric error bars that would yield results that are out of range (e.g. negative error rates).
- If error bars are reported in tables or plots, The authors should explain in the text how they were calculated and reference the corresponding figures or tables in the text.

8. **Experiments compute resources**

Question: For each experiment, does the paper provide sufficient information on the computer resources (type of compute workers, memory, time of execution) needed to reproduce the experiments?

Answer: [Yes]

Justification: The experiments consist primarily of synthetic simulations and closed-form estimators, which are computationally lightweight and do not require specialized hardware. No GPU is needed, and all experiments can be run on a standard laptop or CPU-based machine. While exact runtimes are not reported, the simplicity and efficiency of the estimators ensure that the results are reproducible without significant computational resources.

Guidelines:

- The answer NA means that the paper does not include experiments.
- The paper should indicate the type of compute workers CPU or GPU, internal cluster, or cloud provider, including relevant memory and storage.
- The paper should provide the amount of compute required for each of the individual experimental runs as well as estimate the total compute.
- The paper should disclose whether the full research project required more compute than the experiments reported in the paper (e.g., preliminary or failed experiments that didn't make it into the paper).

9. **Code of ethics**

Question: Does the research conducted in the paper conform, in every respect, with the NeurIPS Code of Ethics https://neurips.cc/public/EthicsGuidelines?

Answer: [Yes]

Justification: The research adheres fully to the NeurIPS Code of Ethics. The work is theoretical and empirical in nature, with experiments conducted on synthetic data and the publicly available CIFAR-10 dataset. No personal, sensitive, or proprietary data is used. All methods are described transparently, reproducibility is supported through open supplementary materials, and no foreseeable negative societal impacts or misuse of the research have been identified.

Guidelines:

- The answer NA means that the authors have not reviewed the NeurIPS Code of Ethics.
- If the authors answer No, they should explain the special circumstances that require a deviation from the Code of Ethics.
- The authors should make sure to preserve anonymity (e.g., if there is a special consideration due to laws or regulations in their jurisdiction).

10. **Broader impacts**

Question: Does the paper discuss both potential positive societal impacts and negative societal impacts of the work performed?

Answer: [NA]

Justification: No dedicated section is included in the paper to discuss societal impacts. However, the work is theoretical in nature and focuses on a methodological contribution to matrix-valued regression. As such, it does not directly target any high-risk application domains. While the methods may have positive downstream impact in areas such as spatiotemporal modeling or image analysis, no immediate or foreseeable negative societal impacts are associated with the research.

Guidelines:

- The answer NA means that there is no societal impact of the work performed.
- If the authors answer NA or No, they should explain why their work has no societal impact or why the paper does not address societal impact.
- Examples of negative societal impacts include potential malicious or unintended uses (e.g., disinformation, generating fake profiles, surveillance), fairness considerations (e.g., deployment of technologies that could make decisions that unfairly impact specific groups), privacy considerations, and security considerations.
- The conference expects that many papers will be foundational research and not tied to particular applications, let alone deployments. However, if there is a direct path to any negative applications, the authors should point it out. For example, it is legitimate to point out that an improvement in the quality of generative models could be used to generate deepfakes for disinformation. On the other hand, it is not needed to point out that a generic algorithm for optimizing neural networks could enable people to train models that generate Deepfakes faster.
- The authors should consider possible harms that could arise when the technology is being used as intended and functioning correctly, harms that could arise when the technology is being used as intended but gives incorrect results, and harms following from (intentional or unintentional) misuse of the technology.

- If there are negative societal impacts, the authors could also discuss possible mitigation strategies (e.g., gated release of models, providing defenses in addition to attacks, mechanisms for monitoring misuse, mechanisms to monitor how a system learns from feedback over time, improving the efficiency and accessibility of ML).

11. **Safeguards**

   Question: Does the paper describe safeguards that have been put in place for responsible release of data or models that have a high risk for misuse (e.g., pretrained language models, image generators, or scraped datasets)?

   Answer: [NA]

   Justification: The paper does not involve the release of high-risk models or datasets. It focuses on theoretical analysis and simulation-based evaluation, along with experiments on the publicly available CIFAR-10 dataset. No pretrained models, sensitive data, or generative systems are used.

   Guidelines:

   - The answer NA means that the paper poses no such risks.
   - Released models that have a high risk for misuse or dual-use should be released with necessary safeguards to allow for controlled use of the model, for example by requiring that users adhere to usage guidelines or restrictions to access the model or implementing safety filters.
   - Datasets that have been scraped from the Internet could pose safety risks. The authors should describe how they avoided releasing unsafe images.
   - We recognize that providing effective safeguards is challenging, and many papers do not require this, but we encourage authors to take this into account and make a best faith effort.

12. **Licenses for existing assets**

   Question: Are the creators or original owners of assets (e.g., code, data, models), used in the paper, properly credited and are the license and terms of use explicitly mentioned and properly respected?

   Answer: [Yes]

   Justification: All external assets used in the paper are publicly available and properly credited. The CIFAR-10 dataset is cited appropriately and used in accordance with its terms of use. Any third-party libraries or tools used for simulations or experiments (e.g., NumPy, TensorFlow, scikit-learn) follow open-source licenses and are acknowledged either directly in the text or within the supplementary notebook. No proprietary or restricted-use resources are employed.

   Guidelines:

   - The answer NA means that the paper does not use existing assets.
   - The authors should cite the original paper that produced the code package or dataset.
   - The authors should state which version of the asset is used and, if possible, include a URL.
   - The name of the license (e.g., CC-BY 4.0) should be included for each asset.
   - For scraped data from a particular source (e.g., website), the copyright and terms of service of that source should be provided.
   - If assets are released, the license, copyright information, and terms of use in the package should be provided. For popular datasets, `paperswithcode.com/datasets` has curated licenses for some datasets. Their licensing guide can help determine the license of a dataset.
   - For existing datasets that are re-packaged, both the original license and the license of the derived asset (if it has changed) should be provided.
   - If this information is not available online, the authors are encouraged to reach out to the asset's creators.

13. **New assets**

Question: Are new assets introduced in the paper well documented and is the documentation provided alongside the assets?

Answer: [Yes]

Justification: The paper introduces new simulation code and experiments, which are provided in a well-documented Jupyter notebook included in the supplementary material. The code includes clear instructions, parameter settings, and explanations necessary to reproduce the results. No new datasets or models involving human subjects or requiring consent are introduced. All assets are anonymized to comply with the double-blind review policy.

Guidelines:

- The answer NA means that the paper does not release new assets.
- Researchers should communicate the details of the dataset/code/model as part of their submissions via structured templates. This includes details about training, license, limitations, etc.
- The paper should discuss whether and how consent was obtained from people whose asset is used.
- At submission time, remember to anonymize your assets (if applicable). You can either create an anonymized URL or include an anonymized zip file.

14. **Crowdsourcing and research with human subjects**

Question: For crowdsourcing experiments and research with human subjects, does the paper include the full text of instructions given to participants and screenshots, if applicable, as well as details about compensation (if any)?

Answer: [NA]

Justification: The paper does not involve crowdsourcing or research with human subjects. All experiments are conducted on synthetic data and the publicly available CIFAR-10 dataset, which does not contain personally identifiable information or require participant interaction.

Guidelines:

- The answer NA means that the paper does not involve crowdsourcing nor research with human subjects.
- Including this information in the supplemental material is fine, but if the main contribution of the paper involves human subjects, then as much detail as possible should be included in the main paper.
- According to the NeurIPS Code of Ethics, workers involved in data collection, curation, or other labor should be paid at least the minimum wage in the country of the data collector.

15. **Institutional review board (IRB) approvals or equivalent for research with human subjects**

Question: Does the paper describe potential risks incurred by study participants, whether such risks were disclosed to the subjects, and whether Institutional Review Board (IRB) approvals (or an equivalent approval/review based on the requirements of your country or institution) were obtained?

Answer: [NA]

Justification: The paper does not involve human subjects or study participants. All experiments are conducted using synthetic data or the CIFAR-10 dataset, which is publicly available and does not involve any human interaction or identifiable information. Therefore, no IRB approval was required.

Guidelines:

- The answer NA means that the paper does not involve crowdsourcing nor research with human subjects.
- Depending on the country in which research is conducted, IRB approval (or equivalent) may be required for any human subjects research. If you obtained IRB approval, you should clearly state this in the paper.

- We recognize that the procedures for this may vary significantly between institutions and locations, and we expect authors to adhere to the NeurIPS Code of Ethics and the guidelines for their institution.
- For initial submissions, do not include any information that would break anonymity (if applicable), such as the institution conducting the review.

16. **Declaration of LLM usage**

Question: Does the paper describe the usage of LLMs if it is an important, original, or non-standard component of the core methods in this research? Note that if the LLM is used only for writing, editing, or formatting purposes and does not impact the core methodology, scientific rigorousness, or originality of the research, declaration is not required.

Answer: [NA]

Justification: The core methods, theoretical analysis, and experimental components of the paper do not involve the use of large language models (LLMs) in any important, original, or non-standard way. Any use of LLMs was limited to editing support and does not affect the scientific contributions of the work.

Guidelines:

- The answer NA means that the core method development in this research does not involve LLMs as any important, original, or non-standard components.
- Please refer to our LLM policy (`https://neurips.cc/Conferences/2025/LLM`) for what should or should not be described.

