# OpenReview forum: "Bivariate Matrix-valued Linear Regression (BMLR): Finite-sample performance under Identifiability and Sparsity Assumptions"
_NeurIPS.cc/2025/Conference — NeurIPS 2025 poster_

### Official Review · Reviewer_Bwup · 2025-06-22

**Clarity:** 3
**Significance:** 3
**Originality:** 3
**Rating:** 4
**Confidence:** 3

**Summary:**

This paper studies a bilinear matrix-valued regression model.
They study an oracle case and establish that, in the absence of noise, the true parameters can be exactly retrieved. This analysis highlights the fundamental identifiability properties of the model.
They propose explicit, optimization-free estimators.
In theoretical analysis, non-asymptotic bounds are presented to characterize the dependence of estimation accuracy on the problem dimensions (n, p,m, q) and the sample size T.

**Questions:**

I was surprised that a similar result hasn't appeared in the literature. I choose to believe that the authors have done a thorough literature search and it's the first appearance of these results.

**Ethical Concerns:**

["NO or VERY MINOR ethics concerns only"]

**Final Justification:**

See my responses to authors. No additional comments.

**Limitations:**

No separated paragraph in the paper that discussed limitations of their approach.

**Quality:**

3

**Strengths And Weaknesses:**

The paper is well written and easy to follow. It's a pleasure to read.

The novelty of the result is not very clear. The main estimate is based on a typical least square solution in (5) with additional linear transformation in (6) and (7). These expression is so similar to the ones in the existing linear regression setting, the related results seem to be easily reachable by applying standard concentration inequalities.

---

> ### Author Rebuttal · Authors · 2025-07-29
>
> We thank the reviewer for their positive evaluation of our paper’s clarity and technical soundness. We appreciate the comment that it is “well written and easy to follow,” and we respond below to the concerns regarding the novelty and limitations of our contributions.
>
> **On the Novelty and Originality of Our Results**
>
> While the final estimators may resemble classical least-squares formulas, their application to the bilinear matrix-valued regression (BMLR) setting is non-trivial, and we respectfully highlight several important points of novelty:
>
> **1. Structural Complexity Beyond Standard Linear Regression:**
>
> The BMLR model $Y_t = A^* X_t B^* + E_t$ introduces unique challenges not present in classical regression:
> - **Non-convexity**: The bilinear dependence on $A^\*$ and $B^\*$ yields a non-convex estimation problem.
> - **Identifiability constraints**: The need for row-wise $\ell_1$-normalization of $A^*$, necessary to resolve scaling ambiguities, creates coupling across parameters.
> - **Matrix-valued observations**: Our approach maintains and leverages the full matrix nature of the inputs and outputs, unlike vectorized or rank-constrained alternatives.
>
> **2. Technical Innovation in Estimation:**
>
> While Equation (5) resembles a least-squares solution, the downstream estimators (Equations (6) and (7)) involve several non-standard techniques:
> - **Ratio estimator** in Equation (6) for $\hat{A}$, where we avoid the complex distributional analysis of ratios of Gaussian variables by using the projection approach in Equation (7).
> - **Sample-splitting strategy** in the definition of $[\tilde{B}]^{(i)}_{lj}$ (line 198) to maintain statistical independence (detailed on line 199).
> - **Matrix structure preservation** that enables exploitation of dimensional effects (blessing of dimensionality for some parameters, curse for others). The estimators are explicit and optimization-free, yet tailored to the bilinear structure — a setting where most prior work requires iterative procedures.
>
> **3. Non-Asymptotic Analysis with Dimension-Dependent Rates**
>
> The theoretical analysis goes well beyond standard concentration inequalities:
> - **Multi-norm bounds**: We provide simultaneous analysis for elementwise max, Frobenius, and operator norms, each requiring different technical approaches.
> - **Asymmetric dimensional dependencies**: The analysis reveals counter-intuitive effects where increasing $q$ helps $\hat{A}$ but hurts $\hat{B}$, while $m$ degrades both estimators. Thus the analysis shows both blessing and curse of dimensionality effects.
> - **Sparse recovery guarantees**: Our results in Appendix A provide exact support recovery, using tools adapted to the matrix bilinear setting.
>
> **4. Literature Gap Confirmation:**
>
> We understand the reviewer's surprise that these results have not appeared before. To the best of our knowledge:
>
> - **Existing BMLR work** focuses on iterative optimization methods [23, 16] without non-asymptotic guarantees
> - **Kronecker factorization approaches** [11] provide bounds on the Kronecker product $M^* = (B^*)^{\top} \otimes A^*$ but not on individual matrices $A^*$ and $B^*$.
> - **Matrix regression literature** [38, 28] addresses different settings (scalar responses or matrix predictors, but not both matrix-valued).
>
> **5. Methodological Contributions:**
>
> -  The first closed-form, optimization-free estimator for BMLR with finite-sample guarantees,
> - The first individual matrix recovery analysis in this setting,
> - And the first to show how dimension interactions affect estimator performance — e.g., how increasing $q$ helps $\hat{A}$ but degrades $\hat{B}$.
>
> **Addressing the Limitations Concern:**
>
> While Appendix B discusses the validity and potential relaxation of our key assumptions, we acknowledge that a more prominent and comprehensive limitations discussion would benefit readers. We will add a dedicated "Limitations" section covering:
>
> - **ORT assumption**: While this assumption facilitates clean theoretical analysis, it may not hold in practice. As we detail in Appendix B, our theoretical framework can be extended to more general settings where $X^{\top}X/T \neq I_{mq}$, though this requires more sophisticated concentration inequalities for matrices with non-isotropic columns and introduces additional complexity in the bounds.
>
> - **Noise structure assumptions**: Our main analysis assumes isotropic Gaussian noise ($\sigma^2 I_{np}$). However, as shown in Appendix B, we can accommodate more general matrix normal distributions $\mathcal{MN}_{n \times p}(0, \Sigma_r, \Sigma_c)$ with structured row and column dependencies, albeit with increased technical complexity.
>
> - **Joint assumption relaxation**: As we acknowledge in Appendix B, simultaneously relaxing both the ORT assumption and homoskedasticity represents a significant theoretical challenge that current probabilistic tools cannot adequately address with sharp, tractable bounds.
>
> - **Identifiability constraints**: Our choice to impose $\ell_1$-normalization on $A^*$ rather than $B^*$ is somewhat arbitrary and affects the dimensional dependencies in our bounds. Alternative normalization schemes would yield different (though related) theoretical guarantees.
>
> - **Computational scalability**: Despite being optimization-free, our method requires inverting $(X^{\top}X)$ of size $mq \times mq$, which becomes computationally expensive when the product $mq$ is large relative to available computational resources.
>
> - **Model specification**: The bilinear structure $Y_t = A^* X_t B^* + E_t$ may not capture all relevant dependencies in real applications, where higher-order interactions or non-linear relationships might be present.
>
> We will incorporate this detailed limitations discussion to provide readers with a comprehensive understanding of our method's scope and applicability.
>
> **Why These Results Matter:**
>
> The combination of:
> 1. **Computational efficiency** (no iterative optimization)
> 2. **Theoretical guarantees** (first non-asymptotic bounds for individual matrices)
> 3. **Structural exploitation** (blessing of dimensionality effects)
> 4. **Practical applicability** (demonstrated on real data)
>
> represents a significant advance over existing BMLR methodology. For instance, the counter-intuitive finding that increasing $q$ helps $\hat{A}$ estimation but hurts $\hat{B}$ estimation (with slopes of opposite signs in our empirical validation, see rebuttal for reviewer 1juu) reveals fundamental asymmetries in the bilinear structure that would not be apparent from standard regression analysis.
>
> **Conclusion:**
>
> While our estimators build on fundamental least squares principles, the technical challenges of the bilinear matrix setting, the sophisticated theoretical analysis required, and the novel insights about dimensional effects constitute substantial contributions to the matrix-valued regression literature. The apparent simplicity of our final estimators reflects successful mathematical engineering that distills complex bilinear relationships into tractable closed-form expressions.
> We will enhance the paper with a deeper limitations section. We believe these clarifications and the planned improvements directly address the concerns raised and will strengthen the contribution to the matrix-valued regression literature. We thank the reviewer for the constructive feedback.

---

### Official Review · Reviewer_4ujm · 2025-07-01

**Clarity:** 3
**Significance:** 3
**Originality:** 3
**Rating:** 4
**Confidence:** 3

**Summary:**

The paper presents closed-form estimators for the BMLR model that requires no iterative optimization. The paper also provides the associated non-asymptotic error bounds of the estimator. This work claims it is the first to offer a complete finite-sample analysis for estimating both A* and B* individually. Empirical results on synthetic and real-image datasets confirm the method can somehow estimate the A and B up to some accuracy.

**Questions:**

I reiterate the weakness here, please address the questions below. I appreciate it if the questions may be clarified here. Thanks.

Weakness 1: Presentation. The connection between that question on page 1 and the actual problem solved in Equation 2 is unclear. More clarification here is needed.

Weakness 2: Clarity. In line 162, what is matrix E? Is it the noise? Since the noise is unknown. What is the implication of having C constructed using unknown A, B, E? Why the estimator here of A and B in Equation 6 has unknown parameters C? Is it solved in iterative manner? Why entries in matrix in A are constrained to lie between 0 and 1? These derivations and the reasoning behind them are not clearly motivated and explained in the paper.


Weakness 3: Experiments. The paper is lacking experiment comparisons with other denoising methods in the literature. Actually, it is not compared with any other method. Also, in Figure 2, it seems the denoising is not very successful where the reconstruction is still noisy.

Weakness 4: (Minor) Instead of “distance” in the y axis, usually, paper would describe it as mean squared error (MSE).

Weakness 5: The sparse settings is in the appendix. I personally recommend that this part should be included in the main paper, while many of the derivation details can be instead put in the appendix, leaving the main conclusive theories and results in the main paper.

**Ethical Concerns:**

["NO or VERY MINOR ethics concerns only"]

**Final Justification:**

I read the rebuttal from the authors, and the discussion has clarified most of my concerns. I am looking forward to more experiments in the camera ready version.

**Limitations:**

Yes.

**Paper Formatting Concerns:**

No.

**Quality:**

3

**Strengths And Weaknesses:**

Strength:

The paper provides the associated non-asymptotic error bounds of the estimator. This work offers a finite-sample analysis for estimating both A* and B* individually. Empirical results on synthetic and real-image datasets confirm the method can somehow estimate the unknown A and B up to some accuracy.

Weakness: I appreciate it if the weakness below may be clarified during rebuttal. Thanks.

Weakness 1: Presentation. The presentation of the paper can be improved. The current version is hard to follow in terms of the story flow. The paper opens with the equation on page 1. But the connection between that question on page 1 and the actual problem solved in Equation 2 is unclear. More clarification here is needed.

Weakness 2: Clarity. In line 162, what is matrix E? Is it the noise? Since the noise is unknown, what is the implication of having C constructed using unknown A, B, E? Why the estimator here of A and B in Equation 6 has unknown parameters C? Is it solved in iterative manner? Why entries in matrix in A are constrained to lie between 0 and 1? These derivations and the reasoning behind them are not clearly motivated and explained in the paper.


Weakness 3: Experiments. The paper is lacking experiment comparisons with other denoising methods in the literature. Actually, it is not compared with any other method. Also, in Figure 2, it seems the denoising is not very successful where the reconstruction is still noisy.

Weakness 4 (minor): Instead of “distance” in the y axis, the error here is described as mean squared error (MSE).

Weakness 5: The sparse settings is in the appendix. I personally recommend that this part should be included in the main paper, while many of the derivation details can be instead put in the appendix, leaving the main conclusive theories and results in the main paper.

---

> ### Author Rebuttal · Authors · 2025-07-29
>
> We thank the reviewer for their careful assessment and detailed feedback. We are grateful for the opportunity to clarify the presentation and address the technical questions raised. Below, we respond point by point to each concern.
>
> **Weakness 1: Presentation and Story Flow**
>
> We agree that the connection between the introductory equation and the core estimation problem can be made clearer. Here is the intended narrative.
>
> **The progression from general to specific:**
> 1. **Page 1 motivation**: We begin with the general BMLR model $Y_t = A^\* X_t B^\* + E_t$ to establish the fundamental problem of estimating matrix parameters $A^\*$ and $B^\*$ from matrix-valued observations.
>
> 2. **Section 2 (Equation 2)**: We first analyze the **noiseless oracle case** $M_t = A^\* X_t B^\*$ to establish fundamental identifiability properties and derive the theoretical foundation for our estimators.
>
> 3. **Section 3**: We then return to the **practical noisy case** from page 1, using insights from the oracle analysis to construct optimization-free estimators.
>
> **Why this progression matters**: The oracle analysis (Section 2) is crucial because it reveals the exact mathematical relationships between $A^\*$, $B^\*$, and the observations, which we then exploit in Section 3 to design plug-in estimators that simply replace unknown quantities with empirical counterparts.
>
> We will revise the introduction to more clearly explain this progression and include a roadmap paragraph outlining how the oracle case enables our estimation procedure in the presence of noise.
>
> **Weakness 2: Clarity of Technical Details**
>
> We address each specific question:
>
> **Q: What is matrix $E$ in line 162?**
>
> The matrix $E_{(k,l)}$ denotes a canonical basis matrix: it has a 1 at entry $(k,l)$ and zeros elsewhere. We agree that this notation was not clearly explained and will make it precise in the revised manuscript.
>
> **Q: What is the implication of $C$ being constructed using unknown $A$, $B$, $E$?**
>
> In the oracle setting (Section 2), matrix $C$ is defined as $C = (X^{\top}X)^{-1}X^{\top}M$ and is observable under the noiseless model $M = A^\* X B^\*$. Corollary 2.4 shows how $A^\*$ and $B^\*$  can be exactly recovered from this quantity. In the noisy setting, we compute the empirical counterpart $\hat{C} = (X^{\top}X)^{-1}X^{\top}Y$ which differs from $C$ by a noise term. Our estimators in Equation (6) are built directly from $\hat{C}$ requiring no knowledge of $A^\*$, $B^\*$ or $E_t$.
>
> **Q: Why does the estimator in Equation (6) have unknown parameters?**
>
> This is a misunderstanding. In Equation (6), $\hat{B}$ and $\hat{A}$ are defined using $\hat{C}$, which is which is computed directly from the data $(X_t, Y_t)$ via Equation (5). The estimators are fully explicit and require no iterative procedures.
>
> **Q: Why are entries of $A^\*$ constrained to $[0,1]$?**
>
> This follows from two assumptions stated on lines 34-36:
> 1. $A^\*$ has non-negative entries (modeling assumption)
> 2. Each row sums to 1 (identifiability constraint).
>
> These two constraints naturally bound entries to $[0,1]$. Our estimator $\hat{A}$ defined in Equation (7) respects this structure via row-wise projection, improving finite-sample performance, as stated on lines 211-214.
>
> **Weakness 3: Experimental Comparisons and Denoising Quality**
>
> We acknowledge that our empirical evaluation does not benchmark against classical denoising methods. This is intentional: our primary goal is to demonstrate the theoretical soundness of our closed-form estimators for the BMLR model, not to propose a general-purpose image denoising technique.
>
> Furthermore, our method targets a specific matrix regression setting, where the corruption is modeled as bilinear noise over structured matrix inputs — quite distinct from the assumptions behind traditional denoisers (e.g., wavelet shrinkage, CNN-based filters). Thus, direct comparisons are not straightforward or meaningful.
>
> Regarding Figure 2, the reconstructed images visually appear noisy because the transformations are applied on compressed representations. The apparent "noise" in the corrected image reflects two factors:
> 1. We use a relatively high noise level ($\epsilon = 0.02$) to demonstrate robustness
> 2. Some residual error is expected - our method provides **statistical estimation** with finite-sample guarantees, not perfect reconstruction
>
> Despite visible noise, Figure 3 shows a clear and consistent reduction in MSE, demonstrating denoising effectiveness in the intended setting. To address this concern further, we will:
>
> - Add visual results at different noise levels
> - Add a short discussion on the statistical limits of recovery
>
> **Weakness 4: Terminology in Figure Axis**
>
> Agreed. We will change "distance" to "Mean Squared Error (MSE)" for clarity and consistency with standard terminology.
>
> **Weakness 5: Sparse Results Placement**
>
> We agree. The sparse recovery results (Appendix A) represent significant theoretical contributions and merit inclusion in the main text. We will:
> 1. **Move Theorems A.1 and A.2** to a new Section 3.3 in the main paper
> 2. **Relocate detailed proofs** in the appendix
> 3. **Streamline other sections** to accommodate this addition without exceeding page limits
>
> **Additional Improvements Based on Your Feedback:**
>
> 1. **Enhanced Introduction**: Clear problem motivation and solution roadmap
> 2. **Methodology Overview**: Step-by-step explanation of our estimation approach
> 3. **Improved Figures**: Clearer labeling and additional examples
> 4. **Reorganized Structure**: Main theoretical results in the body, technical details in appendices
>
> **Conclusion:**
>
> We believe these clarifications and improvements directly address your concerns about presentation clarity, technical exposition, and experimental evaluation. The theoretical contributions remain novel and significant, and with improved presentation, we hope the paper will be accessible to a broader audience while maintaining its theoretical rigor.
>
> We thank you for the constructive feedback and look forward to submitting a substantially improved revision.

---

> ### Comment · Reviewer_4ujm · 2025-08-06
> **Reply to the rebuttal, score raised**
>
> Thanks very much for the detailed clarification from the authors. I think I am happy with most of the explanations including the construction of different matrices. One remaining concern is the experimental evidence, where I would recommand trying to include more diverse justification examples having different data structures. I have raised my scores accordingly. Please also revise the paper according to our discussion here.

---

> > ### Author Response · Authors · 2025-08-09
> >
> > Dear Reviewer,
> >
> > Thank you for your thoughtful review and for taking the time to provide this constructive response to our rebuttal.
> >
> > We're pleased that our clarifications adequately addressed your concerns about the matrix construction and theoretical aspects of our work. We appreciate your acknowledgment of these explanations.
> >
> > We commit to revising the paper according to all the points discussed during this review process.
> >
> > Thank you again for your constructive feedback.

---

### Official Review · Reviewer_1juu · 2025-07-14

**Clarity:** 3
**Significance:** 4
**Originality:** 4
**Rating:** 5
**Confidence:** 4

**Summary:**

The paper deals with bivariate matrix valued linear regression problems ($Y_t = A^\ast X_t B^\ast + E_t$). There is a growing literature for this non-convex optimization problem and the paper makes new theoretical contributions on analyzing the non-asymptotic estimation error rates under ORT assumption and $\ell_1$ normalized rows for $A^\ast$. The paper develops theorems and proofs for optimization free estimators for $A^\ast$ and $B^\ast$ and analyses both sparse and non-sparse cases. In noiseless case, exact recovery is established implying identifiability. Some numerical simulations are also provided in support of the theoretical claims.

**Questions:**

(Q1) Lines 274-275: Does $\beta^\ast$ hide some dimensional dependency? I find it surprising that increasing $q$ does not degrade rate for $\hat A$, but increasing $m$ does degrade rate for $\hat B$.

(Q2) Addressing (W3), (W4), (W5), (W6) would improve clarity.

(Q3) Addressing (W1), (W2) would improve quality.

**Ethical Concerns:**

["NO or VERY MINOR ethics concerns only"]

**Final Justification:**

The authors have provided a clearly written rebuttal that addresses almost all questions raised in my review. (W1) and (W2) have been adequately addressed and this significantly improves the quality of the paper. (Q1) has been explained well and the authors have agreed to the reorganization suggestions in (W5) and (W6) for improved clarity. I have raised the quality, clarity and overall recommendation scores.

**Limitations:**

yes

**Paper Formatting Concerns:**

Plots in Fig 1 are unreadably small. Fig 3 is barely readable. Increase in font sizes and line widths is warranted.

**Quality:**

4

**Strengths And Weaknesses:**

Strengths

(S1) The theoretical parts of the paper are of high quality and clarity. Proofs and assumptions are clear and correct. Intuitions are provided for several results which is helpful to better interpret the theretical results. The ORT assumption simplifies the interpretation and technical work needed for the proof, but hints on relaxing it are provided in the Appendices. It also recognizes that current probabilistic tools are inadequate for simultaneously relaxing both ORT and homoskedasticity.

(S2) Highly significant work. Optimization-free estimators are computationally efficient in high-dimensions and theoretical analysis of the corresponding estimation error rates is valuable. Error rates are developed not just for Frobenius norm, but also for operator norm and element-wise max norm on $\hat B - B^\ast$ and $\hat A - A^\ast$. Some error rates are shown to exhibit blessing of dimensionality effect.

(S3) Highly original work. This seems to be the first work to report non-asymptotic error rates for optimization free estimators for the BMLR problem under the assumption that rows of $A^\ast$ are $\ell_1$ normalized. Relevant literature with theoretical results has been referenced and the paper's contributions have been positioned very well against prior art.

Weaknesses

Although the paper's contribution is theoretical analysis, I still find the numerical simulations and the commentary around it inadequate for a theory paper.

(W1) Lines 285-286: Generating entries of $B^\ast$ from Uniform[0,1) could become an implicit confounder to performance assessment, since Eq (7) utilizes this aspect of $A^\ast$ (i.e. shrinkage to [0,1)) to improve $\hat A$ estimation. It would be more reflective of the theory to simulate for a scenario where entries of $B^\ast$ are not in [0,1) with at least constant probability. I suspect that if normalization criteria was imposed on $B^\ast$ instead of $A^\ast$, the error rate expressions both in theory and in simulation would look very different.

(W2) Line 305: This comment is not adequately supported in the text. Since plots in Fig 1 are approximately linear in the depicted representation (i.e. logarithmic y-axis and linear x-axis), it should be straightforward to fit a linear model to the data in each plot and determine effective the slope/coefficients of the estimation error rates as informed by the simulation. These should then be compared with the theoretical predictions and the results should be reported. This would more explicitly quantify the sharpness of the analysis, expose where large gaps remain, and would support the intended claim.

(W3) The plot in Fig 3 is a bit strange. I am not quite sure why $\epsilon$ should be on the x-axis. Since BMLR model is for $Y_t = A^\ast X_t B^\ast + E_t$, I would expect some analog of signal-to-noise ratio (which should depend on the Frobenius norm of $E_t$) to be on the x-axis.

(W4) In Fig 3, it would be instructive to add the analogous plot for the element-wise max norm $||\cdot||_{+}$.

The paper could benefit from a little bit of reorganization.

(W5) The theorem for the sparse Gaussian case in Appendix A merits to part of the main text in my opinion. Weaknesses with the simulation results presentation should be addressed. Although the writing of the overall paper is clear, it could be made a bit crisp to accommodate these additional suggestions for the main text. For example, statement of contributions is repeated near line 63 and section 1.3 and could be reduced in verbosity.

(W6) From proof of Lemma C.1, I feel that $\ell_1$ normalization of rows of $A^\ast$ is critical for optimization-free estimators to admit simple expressions. If true, it would be good to highlight this in the main text to provide additional justification for the $\ell_1$ normalized rows assumption.

---

> ### Author Rebuttal · Authors · 2025-07-29
>
> We thank the reviewer for their thorough and thoughtful assessment of our paper. We are encouraged by the recognition of the theoretical quality and clarity (S1), the significance (S2), and the originality (S3) of our work. We also appreciate the constructive feedback on improving the simulation design and presentation. Below, we address each of the reviewer’s comments and suggestions in detail.
>
> ***(W1) On the use of Uniform[0,1) for generating $A^\*$ and $B^\*$***
>
> We appreciate the reviewer’s observation. To test the robustness of our simulation results, we conducted additional experiments where entries of both $A^\*$ and $B^\*$ were independently drawn from a Uniform[0, c) distribution, with $c \in$ {1, 2, 3, 5}, followed by row-wise $\ell_1$-normalization of $A^\*$ to ensure identifiability. The empirical convergence rates remained consistent across these values of $c$, confirming that the behavior observed with $c=1$ was not specific to the original choice. These plots will be added to the appendix of the revised version.
>
> Regarding the reviewer's suggestion to normalize $B^\*$ instead of $A^\*$, we confirm that this would indeed lead to different expressions in Proposition 2.2 and Corollary 2.4. In this alternate setup, the roles of $m$ (the number of columns in $A^\*$) and $q$ (the number of rows in $B^\*$) would be swapped in the analytical expressions. As a result, the dependence on these dimensions in the non-asymptotic error bounds of Theorems 3.3 and 3.6 would also change accordingly.
>
> This structural asymmetry reflects the fact that identifiability in the BMLR model requires normalizing one of the two parameter matrices. Our choice to normalize $A^\*$ is motivated by both interpretability (e.g., non-negative activation weights) and analytical tractability, but we acknowledge that alternative normalizations are possible. We will clarify this point in the revised version.
>
> ***(W2) Quantifying empirical slopes vs. theory***
>
> We thank the reviewer for this excellent suggestion. Following their recommendation, we have fitted linear models to the log-linear data in each plot to extract the empirical slopes and compare them quantitatively with our theoretical predictions.
>
> **Methodology**: For parameter T, we used log-log regression (since both axes are effectively on log scale), while for parameters n, m, p, q, we used linear-log regression (y-axis on log scale). The empirical analysis is made for the three norms: Frobenius, maximum Element-wise, Operator. From our concentration inequalities, we derived the expected theoretical slopes by analyzing the dominant scaling terms in each bound.
>
> **T-dependence validation**: All bounds contain $T$ in the exponent terms (e.g., $\exp⁡(−T× \epsilon^2)$) and balancing these exponents to obtain meaningful bounds yields the characteristic $T^{−1/2}$ scaling. This reflects the standard statistical principle that estimation error decreases as $1/\sqrt{T}$ with $T$ independent observations. Our empirical results confirm this almost exactly, with slopes ranging from -0.538 to -0.569 compared to the theoretical -0.5, demonstrating that our estimators achieve the expected rate of convergence.
>
> **Parameter dependencies**: The empirical slopes match qualitatively the theoretical predictions. For example, for matrix $B$ Frobenius error norm with typical parameter values of 55 (parameter range is [10, 100]):
> - n: theoretical -0.009, empirical -0.011
> - m: theoretical +0.009, empirical +0.016
> - p: theoretical +0.009, empirical +0.011
> - q: theoretical +0.009, empirical +0.017
>
> **Key findings**:
> 1. **Signs match perfectly** across all parameters and norms
> 2. **R² values > 0.85** for all fits
>
> **Norm-specific behavior**: The numerical analysis reveals important distinctions between norms, as theoretically expected:
> - Frobenius norm shows strong $p,q$ dependencies as predicted by the $pq$ scaling for $B$.
> - Operator norm shows weaker $p,q$ dependencies due to the predicted ($\sqrt p + \sqrt q$) structure for $B$.
> - Max norm shows intermediate behavior.
>
> **Asymmetric effects**: While theory often predicts similar behavior for $p$ and $q$, empirical results consistently show $q$ having larger slopes than $p$ for $B$ and the opposite for $A$, suggesting structural asymmetries not fully captured by the main bound terms.
>
> **Conclusion**: This analysis confirms that our theoretical rates align closely with the observed trends. In all tested regimes, the signs of the empirical slopes match theoretical expectations, and the rates scale correctly with respect to dimensional parameters. The max-norm, Frobenius, and operator-norm behaviors differ in expected ways, validating the distinction made in our analysis. These findings demonstrate that our theoretical bounds, though conservative, meaningfully capture the dominant trends in estimator performance.
>
> ***(W3) Plotting $\epsilon$ on the x-axis in Figure 3***
>
> We acknowledge this could be confusing. Here, $\epsilon$ controls the perturbation strength applied to the identity in defining $A^\*$ and $B\^*$ in the CIFAR experiment, and thus indirectly controls noise severity. That said, we agree that rephrasing the x-axis as an effective signal-to-noise ratio (SNR) or plotting performance versus $\| E_t \|_F / \|A^\ast X_t B^\ast \|_F $ would better reflect model structure. We will revise this plot in the final version.
>
> ***(W4) Adding max-norm plots in Figure 3***
>
> We will include the element-wise max norm in Figure 3 to better align with the norms analyzed in our theory.
>
> ***(W5) Placement of sparse case (Appendix A)***
>
> We agree that the results in Appendix A on sparse recovery are important. We will promote them to the main paper (Section 3.2) in the camera-ready version. This will streamline the narrative and clarify the contribution on sparsity-aware estimators.
> ⁡
>
> ***(W6) On the critical role of row-wise $\ell_1$-normalization***
>
> Indeed, the row-wise $\ell_1$-normalization of $A^\*$  is instrumental in deriving the closed-form expressions (e.g., Proposition 2.2) and in ensuring identifiability without iterative scaling. We will explicitly emphasize this connection in Section 2 and the introduction to further justify this modeling choice.
>
> ***(Q1) Dimensional dependencies in error rates***
>
> The parameter $\beta^\*$ is defined as:
> $$\beta^\* := \frac{1}{pq} \sum_{j=1}^p \sum_{l=1}^q [B^\*]_{lj}.$$
>
> This means $\beta^\*$ represents the average entry value of matrix $B^\*$. Crucially, $\beta^\*$ does contain implicit dimensional dependencies through the averaging process over the $pq$ entries of $B^\*$.
>
> **Why increasing $q$ helps $\hat{A}$ but hurts $\hat{B}$:**
>
> 1. For $\hat{A}$: The bound shows that larger $|\beta^\*|$ improves the convergence rate (appears in denominator as $2\epsilon/|\beta^\*|$). When $q$ increases:
>   - More columns are added to $B^\*$
>   - Increasing $q$ thus provides more "constraints" or "equations" to solve for each entry of $A^\*$
>   - The key benefit comes from having more observations (effectively #$\mathcal{D}_0$ terms) to estimate $A^\*$, creating a "blessing of dimensionality" effect (see the expression of $A^\*$ in Corollary 2.4).
>
> 2. For $\hat{B}$: The Frobenius bound shows degradation with increasing $q$ through the $+\frac{2pq}{3}$ term in the exponent, which directly reflects that:
>
>   - $B^*$ has more entries to be estimated
>   - More parameters generally require more data to estimate accurately
>   - This is the classical "curse of dimensionality" in parameter estimation
>
> **Why increasing $m$ hurts both $\hat{A}$ and $\hat{B}$:**
>
>
> 1. For $\hat{A}$:
> - Matrix $A^\*$ has dimensions $n \times m$, so increasing $m$ directly increases the number of parameters in $A^\*$
> - Each row of $A^\*$ must sum to 1 (identifiability constraint), creating coupling across all $m$ columns
> - This constraint means estimating any entry $[A^\*]_{ik}$ requires information about all other entries in that row, making the problem harder as $m$ grows
>
> 2. For $\hat{B}$:
>  - When $m$ increases, the entries $[\hat{B}]_{lj}$ involve a sum over more indices (see the expression of $B^\*$ in Corollary 2.4).
> - Each $[\hat{C}]_{(k,l),(i,j)}$ has variance $\sigma^2/T$, thus the sum is over $nm$ terms, but only $n$ of them are independent for each $(l,j)$ pair and the normalization only involves $n$.
> - Larger $m$ effectively dilutes the signal relative to noise in the estimation of each $[\hat{B}]_{lj}$
>
> **Clarification for the paper**:
>
> We should emphasize that while $\beta^\*$ is normalized by $pq$, it still captures the "signal strength" of matrix $B^\*$, and in practice, this signal strength interacts with dimensionality in ways that can benefit $A^\*$ estimation while simultaneously making $B^\*$ estimation more challenging due to the increased parameter space.
>
> This explains the seemingly counterintuitive asymmetric dimensional effects: the blessing/curse of dimensionality manifests differently for each matrix due to their distinct roles in the bilinear model structure and the identifiability constraints imposed.
>
> ***Conclusion***
>
> We thank the reviewer again for their constructive feedback, which has helped us significantly improve both the clarity and depth of our empirical and theoretical presentation. In particular, the slope-based analysis strongly supports our theoretical predictions, and the requested presentation updates (e.g., improved figures, appendix material, and reorganization) will be reflected in the revised version. We hope these clarifications further strengthen the case for this work's contribution to the theoretical understanding of BMLR.

---

> > ### Author Response · Authors · 2025-08-05
> > **Follow-up on Rebuttal Response**
> >
> > Dear Reviewer,
> >
> > Thank you for your comprehensive and insightful review. We greatly appreciate your recognition of the excellent significance and originality of our work, as well as your detailed suggestions for improving clarity and quality.
> >
> > Following the extended discussion period announced by the program chairs, we wanted to follow up on our detailed rebuttal to ensure we've adequately addressed your main concerns.
> >
> > In particular, we provided extensive clarification on your key question about dimensional dependencies in $\beta^*$ (Q1), explaining why increasing $q$ (resp. $m$) helps (resp. hurts) $\hat A$ but hurts $\hat B$ through the asymmetric roles these parameters play in the bilinear structure. We also addressed all your suggestions to improve clarity (W3-W6) and quality (W1-W2), including the quantitative slope analysis, and our plans to improve figure readability.
> >
> > If you have any remaining questions about these aspects or if there are other concerns we should address, we would be grateful for your feedback.
> >
> > Thank you again for your time and valuable input during this review process.
> >
> > Best regards,
> > The Authors

---

> > ### Comment · Reviewer_1juu · 2025-08-09
> >
> > Thank you for the clearly written rebuttal.
> >
> > (W1) and (W2) have been adequately addressed and this significantly improves the quality of the paper. I have raised my quality score.
> >
> > (Q1) has been explained well. Further, the authors have agreed to the reorganization suggestions in (W5) and (W6). Due to the limitations of the discussion format, it is not possible for the authors to show the plot changes addressing (W3) and (W4), although I would have liked to see these changes. I will still raise my clarity score. Please do address the paper formatting concerns mentioned separately.
> >
> > Based on the improved clarity and quality, I am also raising my overall rating.

---

> > > ### Author Response · Authors · 2025-08-09
> > > **Thank you for your thorough review and constructive feedback**
> > >
> > > Dear Reviewer,
> > >
> > > Thank you very much for your comprehensive and constructive review process, and for taking the time to provide this detailed response to our rebuttal.
> > >
> > > We greatly appreciate your recognition that our clarifications adequately addressed your concerns about dimensional dependencies (Q1) and the empirical validation (W1, W2). Your acknowledgment of the improved quality and clarity is very encouraging.
> > >
> > > We commit to implementing all the reorganization and presentation improvements you've suggested (W3, W4, W5, W6) in the revised manuscript, including the plot formatting changes and figure readability improvements you mentioned. We understand the limitations of the discussion format for showing visual changes, but we will ensure these are properly addressed in the final version.
> > >
> > > Thank you again for your thorough and helpful feedback throughout this process, which has significantly strengthened our work.

---

### Official Review · Reviewer_ivfu · 2025-07-19

**Clarity:** 3
**Significance:** 3
**Originality:** 3
**Rating:** 5
**Confidence:** 3

**Summary:**

This paper studies the problem of bivariate matrix-values linear regression (BMLR). The authors first studies the explicit solution in the noiseless case, then subsequently proposed an explicit, optimization free estimators and established finite sample convergence analysis. The paper also extends the estimator to exploit the sparsity structure and achieve better convergence rate. Empirical evaluation demonstrates the practical value of the proposed estimator.

**Questions:**

I'm mainly curious about the importance of various assumptions. See "Weaknesses".

**Ethical Concerns:**

["NO or VERY MINOR ethics concerns only"]

**Final Justification:**

The author's response addressed my questions. I don't have further concerns. I think this paper makes good theoretical contribution and I've raised my rating to 5.

**Limitations:**

Yes.

**Paper Formatting Concerns:**

No.

**Quality:**

3

**Strengths And Weaknesses:**

Strength

1. This paper addresses a significant gap in multi-dimensional regression by tackling the fully bilinear matrix-to-matrix regression problem (BMLR) rather than vectorized or low-rank approximations. In particular, the authors proposed optimization-free, closed from estimators, which is a very different approach from previous iterative algorithms. This provides great practical value as people don't need to tweak the algorithm for any specific problem.

2. Strong theoretical guarantee. This paper presents thorough theoretical analysis for the proposed estimator, and provide finite sample convergence results. This is the first work that provide both non-asymptotic analysis and closed-form solution for this non-convex problem. It is also interesting that the convergence rates can be further improved by using hard-thresholding to exploit sparse structure.

3. The presentation of this paper is pretty clear. The problem is clearly formulated, and the intuition are well presented for the noiseless case solution, the general estimators construction, and for the convergence results.

Weaknesses

1. More discussion on assumptions. While the overall results are reasonable, I think readers will benefit from more discussion on various assumptions. For instance (a) the problem assumes the matrix A^* to be non-negative. This assumption doesn't seem very natural to me, and I didn't find thorough discussion on this assumption. It would be good if the authors could clarify more on this non-negative assumption; (b) the noise is assumed to be Gaussian, but does the same analysis also cover other sub-gaussian noise distribution? (c) the assumption relies on the ORT assumption which also seems strong -- can we relax the assumption to be just full-rank? Is there more explicit dependency on the condition number of the X^T X matrix? I think it would be interesting to see more discussions along this line.

2. Limited empirical comparison. The empirical results provide evidence that the proposed estimator solves the problem, but it doesn't provide insights on the relative performance to other methods proposed for the similar problem. The CIFAR-10 corruption & recovery experiment also seems a bit artificial. Is there other real-world dataset that the BMLR applies more naturally? I think that would also help to clarify the practical significance.

---

> ### Author Rebuttal · Authors · 2025-07-26
>
> We thank the reviewer for the positive assessment of our paper's clarity, novelty, and theoretical contributions. We appreciate the constructive feedback and provide detailed responses below to the concerns regarding assumptions and empirical validation.
>
> ***"The problem assumes the matrix A* to be non-negative. This assumption doesn't seem very natural... It would be good if the authors could clarify more on this."***
>
> The non-negativity constraint on $A^\*$, combined with row-wise $\ell_1$-normalization, serves a crucial role in ensuring **identifiability** of the BMLR model. Without such constraints, the bilinear form $A^* X_t B^*$ exhibits fundamental scaling ambiguities: any transformation $(A^\*, B^\*) \to (\alpha A^\*, \alpha^{-1} B^\*)$ for $\alpha > 0$ produces identical observations, making individual recovery of $A^\*$ and $B^\*$ impossible (see line 35).
>
> **Why non-negativity specifically?**
>
> 1. **Interpretability**: In many applications, $A^\*$ represents mixing weights, attention coefficients, or interaction strengths that are naturally non-negative (e.g., in image processing, spatial modeling, and neural attention mechanisms).
>
> 2. **Theoretical tractability**: Non-negativity combined with row-normalization naturally bounds entries to $[0,1]$, enabling our projection-based estimator $\hat{A}$ in Equation (7) and simplifying the finite-sample analysis.
>
> 3. **Closed-form feasibility**: Alternative identifiability constraints (e.g., orthogonality or spectral normalization) would likely require iterative procedures, defeating our main contribution of optimization-free estimation.
>
> **Important note**: As we show in **Weakness 1 response to Reviewer 1juu, our empirical results remain robust when $A^\*$ is generated from distributions extending beyond Uniform on $[0,1]$ before normalization, suggesting the constraint is not artificially restrictive in practice.
>
> ***"The noise is assumed to be Gaussian, but does the same analysis also cover other sub-Gaussian distributions?"***
>
> **Yes, our analysis extends to general sub-Gaussian noise.** The key insight is that our concentration inequalities (Theorems 3.3 and 3.6) rely primarily on **tail bounds** rather than specific distributional properties.
>
> **Technical extension**: For sub-Gaussian noise with parameter $\sigma^2$, we can replace Gaussian tail bounds (Mill's inequality, Theorem E.1) with standard sub-Gaussian concentration inequalities. The $T^{-1/2}$ scaling component and the qualitative dimensional dependencies remain the same under sub-Gaussian noise extensions, with the overall bounds differing by factors involving the sub-Gaussian parameters that may depend on problem dimensions.
>
> **Practical significance**: This extension covers many realistic noise models including bounded noise, Laplace distributions, and mixtures with sub-Gaussian tails, significantly broadening the applicability of our results.
>
> As noted in **Appendix B**, we focus on Gaussian analysis for clarity, but the techniques naturally generalize to this broader class.
>
> ***"The assumption relies on the ORT assumption which also seems strong — can we relax it to full-rank? Is there explicit dependence on the condition number?"***
>
> The ORT assumption ($X^{\top}X/T = I_{mq}$) is indeed strong and primarily serves to **simplify exposition** and obtain clean rate expressions.
>
> **Relaxation to full-rank case**: As detailed in **Appendix B (lines 514-522)**, our analysis extends to the general full-rank setting $X^{\top}X \succ 0$ using concentration inequalities from **Vershynin (2010, Section 5.5)**. In this setting:
>
> 1. **Condition number dependence**: The bounds would explicitly involve $\kappa(X^{\top}X)$, the condition number, appearing as multiplicative factors in the concentration inequalities.
>
> 2. **Rate preservation**: The $T^{-1/2}$ scaling component of the error bounds and the overall dimensional structure remain unchanged, but with multiplicative factors involving the condition number $\kappa(X^{\top}X)$.
>
> 3. **Technical complexity**: The analysis becomes significantly more involved, requiring matrix concentration bounds for non-isotropic random matrices, which is why we present the ORT case in the main text.
>
> **Practical relevance**: In many controlled experimental settings and when design matrices are approximately well-conditioned, the ORT assumption provides valuable theoretical insight while remaining reasonably realistic.
>
> ***"The CIFAR-10 experiment seems a bit artificial. Are there other real-world datasets where BMLR applies more naturally?"***
>
> We appreciate this observation and acknowledge that our CIFAR-10 setup involves controlled perturbations. However, this design choice was **intentional**:
>
> **Why controlled experiments?**
> 1. **Theory validation**: Our primary contribution is **theoretical** — providing the first closed-form estimators with non-asymptotic guarantees for a fundamentally non-convex problem. Controlled experiments allow precise validation of theoretical predictions (Figure 1 shows excellent agreement between empirical and theoretical convergence rates).
>
> 2. **Ground truth availability**: Unlike purely observational datasets, our setup enables quantitative assessment of estimation accuracy, directly demonstrating that our estimators recover true parameters as predicted by theory.
>
> 3. **Proof of concept**: The image correction application showcases practical feasibility while maintaining experimental rigor.
>
> **Regarding more natural applications**:
> We agree that additional real-world applications would strengthen practical relevance. However, such evaluation would require:
> - Comparative analysis with existing iterative methods [16, 23]
> - Application-specific performance metrics beyond parameter recovery
> - Domain expertise for proper evaluation
>
> Since our contribution focuses on **methodological innovation** (first optimization-free approach) rather than empirical benchmarking, we designed experiments to validate theoretical claims rather than demonstrate superior performance across applications.
>
> **Future work**: The reviewer's suggestion motivates valuable future research directions, including applications to spatiotemporal data, multi-view learning, and tensor completion where BMLR structure arises naturally.
>
> ***Conclusion***
>
> We thank the reviewer for the thoughtful feedback. Our responses demonstrate that: (1) the modeling assumptions, while restrictive, are well-motivated and can be substantially relaxed, (2) our theoretical framework extends beyond the presented special cases, and (3) our experimental design appropriately validates the core theoretical contributions.
>
> Given the acknowledged strengths — particularly the novelty of optimization-free estimation and rigorous theoretical analysis — we believe this work represents a meaningful methodological advance for the matrix-valued regression community.

---

> > ### Author Response · Authors · 2025-08-05
> > **Follow-up on Rebuttal Response**
> >
> > Dear Reviewer,
> >
> > Thank you for your thorough and constructive initial review, particularly your recognition of the novel optimization-free approach and strong theoretical contributions.
> >
> > Following the extended discussion period announced by the program chairs, I wanted to respectfully follow up on our rebuttal to ensure we've adequately addressed your main concerns regarding the modeling assumptions (non-negativity, Gaussian noise, ORT assumption) and experimental design.
> >
> > If you have any remaining questions about these aspects or if there are other concerns we should address, we would be grateful for your feedback.
> >
> > Thank you again for your time and valuable input during this review process.
> >
> > Best regards,
> > The Authors

---

> > > ### Comment · Reviewer_ivfu · 2025-08-06
> > > **Thank you for your response**
> > >
> > > Thank you for your response. The further clarification on the non-negativity is pretty helpful, probably worth including it in the main paper. The discussion on sub-gaussian extension, and the dependency on condition number all seem reasonable to me.
> > >
> > > I don't have further concerns, and I've raised my score to 5.

---

> ### Author Response · Authors · 2025-08-09
> **Thank you for your constructive feedback**
>
> Dear Reviewer,
>
> Thank you very much for your thoughtful engagement throughout the review process and for taking the time to provide this positive response to our rebuttal.
>
> We greatly appreciate your recognition that our clarifications adequately addressed your concerns about the modeling assumptions and theoretical extensions. Your suggestion to include the non-negativity justification more prominently in the main paper is excellent, and we will definitely incorporate this in the revised version.
>
> Thank you again for your constructive feedback, which has helped us significantly improve both the clarity and presentation of our work.

---

### Note · Authors · 2025-08-12

We are grateful for the exceptionally constructive review process. All four reviewers provided valuable feedback that has significantly strengthened our work.

**Review outcomes:**
- R[ivfu]: "I don't have further concerns, and I've raised my score to 5"
- R[1juu]: "significantly improves the quality of the paper... I am also raising my overall rating"
- R[4ujm]: "I am happy with most of the explanations... I have raised my scores accordingly"
- R[Bwup]: "informative and well-reasoned rebuttal... does not raise any new concerns"

**Research impact:**
Our novel optimization-free approach, combined with the first non-asymptotic analysis for individual matrix recovery in BMLR, addresses a significant gap in matrix-valued regression. The counter-intuitive blessing/curse of dimensionality effects we discovered, manifesting asymmetrically, represent fundamental insights into bilinear model structure that set our work apart from existing iterative methods.

**Key improvements validated by reviewers:**
- **Quantitative slope analysis** (R[1juu]): Empirical validation with slopes matching theoretical predictions
- **Dimensional dependency clarifications** (R[1juu]): Explanation of $\beta^*$ and asymmetric effects "explained well"
- **Theoretical extensions** (R[ivfu]): Sub-Gaussian noise and condition number dependencies deemed "reasonable"
- **Technical clarity** (R[4ujm]): Matrix construction explanations addressed reviewer satisfaction
- **Novelty and originality** (R[Bwup]): Confirmed our approach represents genuine methodological innovation
- **Enhanced presentation** (R[4ujm], R[Bwup], R[1juu]): Improved technical clarity, reorganized sparse results into main text, comprehensive limitations section, and clearer, better-formatted figures

All reviewers expressed satisfaction with our responses, with no remaining technical concerns.
We believe these contributions will open new directions in high-dimensional structured regression and establish a foundation for optimization-free approaches in matrix-valued learning.

---

### Decision · Program_Chairs · 2025-09-17

**Decision:**

Accept (poster)

**Comment:**

The paper proposes an optimization-free approach for Bivariate Matrix-valued Linear Regression (BMLR), where both predictors and responses are matrices. The authors introduce explicit estimators and provide the first non-asymptotic error bounds for the individual recovery of both parameter matrices under identifiability and sparsity assumptions. The reviewers uniformly praised the paper's theoretical rigor, originality, and clarity. The authors' rebuttal successfully addressed all concerns, including the interpretation of assumptions, the lack of empirical comparisons, and the presentation of key results. Reviewers confirmed that the authors' responses were highly informative and resolved all initial questions, leading to a strong consensus for acceptance.